# NEURAL ARCHITECTURE SEARCH OF SPD MANIFOLD NETWORKS

## ABSTRACT

In this paper, we propose a new neural architecture search (NAS) problem of Symmetric Positive Definite (SPD) manifold networks. Unlike the conventional NAS problem, our problem requires to search for a unique computational cell called the SPD cell. This SPD cell serves as a basic building block of SPD neural architectures. An efficient solution to our problem is important to minimize the extraneous manual effort in the SPD neural architecture design. To accomplish this goal, we first introduce a geometrically rich and diverse SPD neural architecture search space for an efficient SPD cell design. Further, we model our new NAS problem using the supernet strategy, which models the architecture search problem as a one-shot training process of a single supernet. Based on the supernet modeling, we exploit a differentiable NAS algorithm on our relaxed continuous search space for SPD neural architecture search. Statistical evaluation of our method on drone, action, and emotion recognition tasks mostly provides better results than the state-of-the-art SPD networks and NAS algorithms. Empirical results show that our algorithm excels in discovering better SPD network design and providing models that are more than 3 times lighter than searched by state-of-the-art NAS algorithms.

## 1 INTRODUCTION

Designing a favorable neural network architecture for a given application requires a lot of time, effort, and domain expertise. To mitigate this issue, researchers in the recent years have started developing algorithms to automate the design process of neural network architectures (Zoph & Le, 2016; Zoph et al., 2018; Liu et al., 2017; 2018a; Real et al., 2019; Liu et al., 2018b; Tian et al., 2020). Although these neural architecture search (NAS) algorithms have shown great potential to provide an optimal architecture for a given application, it is limited to handle architectures with Euclidean operations and representation. To deal with non-euclidean data representation and corresponding set of operations, researchers have barely proposed any NAS algorithms —to the best of our knowledge.

It is well-known that manifold-valued data representation such as symmetric positive definite (SPD) matrices have shown overwhelming accomplishments in many real-world applications such as pedestrian detection (Tuzel et al., 2006; 2008), magnetic resonance imaging analysis (Pennec et al., 2006), action recognition (Harandi et al., 2014), face recognition (Huang et al., 2014; 2015), brain-computer interfaces (Barachant et al., 2011), structure from motion (Kumar et al., 2018; Kumar, 2019), etc. Also, in applications like diffusion tensor imaging of the brain, drone imaging, samples are collected directly as SPD's. As a result, neural network usage based on Euclidean data representation becomes inefficient for those applications. Consequently, this has led to the development of the SPD neural network (SPDNet) architectures for further improvements in these areas of research (Huang & Van Gool, 2017; Brooks et al., 2019). However, these architectures are handcrafted, so the operations or the parameters defined for these networks generally change as per the application. This motivated us to propose a new NAS problem of SPD manifold networks. A solution to this problem can reduce unwanted efforts in SPDNet design. Compared to the traditional NAS problem, our NAS problem requires a new definition of computation cell and proposal for diverse SPD candidate operation set. In particular, we model the basic architecture cell with a specific directed acyclic graph (DAG), where each node is a latent SPD representation, and each edge corresponds to a SPD candidate operation. Here, the intermediate transformations between nodes respect the geometry of the SPD manifolds.

For solving the suggested NAS problem, we exploit a supernet search strategy which models the architecture search problem as a one-shot training process of a supernet that comprises of a mixture

of SPD neural architectures. The supernet modeling enables us to perform a differential architecture search on a continuous relaxation of SPD neural architecture search space, and therefore, can be solved using a gradient descent approach. Our evaluation validates that the proposed method can build a reliable SPD network from scratch. We show the results of our method on benchmark datasets that clearly show results better than handcrafted SPDNet. Our work makes the following contributions:

- We introduce a NAS problem of SPD manifold networks that opens up a new direction of research in automated machine learning and SPD manifold learning. Based on a supernet modeling, we propose a novel differentiable NAS algorithm for SPD neural architecture search. Concretely, we exploit a sparsemax-based Fréchet mixture of SPD operations to introduce sparsity that is essential for an effective diffentiable search, and bi-level optimization with manifold-based update and convexity-based update to jointly optimize architecture parameters and network kernel weights.
- Besides well-studied operations from exiting SPDNets (Huang & Van Gool, 2017; Brooks et al., 2019; Chakraborty et al., 2020), we follow Liu et al. (2018b) to further introduce some new SPD layers, i.e., skip connection, none operation, max pooling and averaging pooling. Our introduced additional set of SPD operations make the search space more diverse for the neural architecture search algorithm to obtain more generalized SPD neural network architectures.
- Evaluation on three benchmark datasets shows that our searched SPD neural architectures can outperform the existing handcrafted SPDNets (Huang & Van Gool, 2017; Brooks et al., 2019; Chakraborty et al., 2020) and the state-of-the-art NAS methods (Liu et al., 2018b; Chu et al., 2020). Notably, our searched architecture is more than 3 times lighter than those searched by the traditional NAS algorithms.

## 2 BACKGROUND

In recent years, plenty of research work has been published in the area of NAS (Gong et al., 2019; Liu et al., 2019; Nayman et al., 2019; Guo et al., 2020). This is probably due to the success of deep learning for several applications which has eventually led to the automation of neural architecture design. Also, improvements in the processing capabilities of machines has influenced the researchers to work out this computationally expensive yet an important problem. Computational cost for some of the well-known NAS algorithms is in thousands of GPU days which has resulted in the development of several computationally efficient methods (Zoph et al., 2018; Real et al., 2019; Liu et al., 2018a; 2017; Baker et al., 2017; Brock et al., 2017; Bender, 2019; Elsken et al., 2017; Cai et al., 2018; Pham et al., 2018; Negrinho & Gordon, 2017; Kandasamy et al., 2018; Chu et al., 2020). In this work, we propose a new NAS problem of SPD networks. We solve this problem using a supernet modeling methodology with a one-shot differentiable training process of an overparameterized supernet. Our modeling is driven by the recent progress in supernet methodology. Supernet methodology has shown a great potential than other NAS methodologies in terms of search efficiency. Since our work is directed towards solving a new NAS problem, we confine our discussion to the work that have greatly influenced our method i.e., one-shot NAS methods and SPD networks.

To the best of our knowledge, there are mainly two types of one-shot NAS methods based on the architecture modeling (Elsken et al., 2018) (a) *parameterized architecture* (Liu et al., 2018b; Zheng et al., 2019; Wu et al., 2019; Chu et al., 2020), and (b) *sampled architecture* (Deb et al., 2002; Chu et al., 2019). In this paper, we adhere to the parametric modeling due to its promising results on conventional neural architectures. A majority of the previous work on NAS with continuous search space fine-tunes the explicit feature of specific architectures (Saxena & Verbeek, 2016; Veniat & Denoyer, 2018; Ahmed & Torresani, 2017; Shin et al., 2018). On the contrary, Liu et al. (2018b); Liang et al. (2019); Zhou et al. (2019); Zhang et al. (2020); Wu et al. (2020); Chu et al. (2020) provides architectural diversity for NAS with highly competitive performances. The other part of our work focuses on SPD network architectures. There exist algorithms to develop handcrafted SPDNet (Huang & Van Gool, 2017; Brooks et al., 2019; Chakraborty et al., 2020). To automate the process of SPD network design, in this work, we choose the most promising approaches from these fields (NAS (Liu et al., 2018b), SPD networks (Huang & Van Gool, 2017)) and propose a NAS algorithm for SPD inputs. Next, we summarize the essential notions of Riemannian geometry of SPD manifolds, followed by an introduction of some basic SPDNet operations and layers. As some of the introduced operations and layers have been well-studied by the existing literature, we applied them directly to define our SPD neural architectures' search space.

**Representation and Operation:** We denote $n \times n$ real SPD as $\boldsymbol{X} \in \mathcal{S}_{++}^n$. A real SPD matrix $\boldsymbol{X} \in \mathcal{S}_{++}^n$ satisfies the property that for any non-zero $z \in \mathbb{R}^n$, $z^T \boldsymbol{X} z > 0$ (Harandi et al., 2017). We denote $\mathcal{T}_{\boldsymbol{X}} \mathcal{M}$ as the tangent space of the manifold $\mathcal{M}$ at $\boldsymbol{X} \in \mathcal{S}_{++}^n$ and $\log$ corresponds to matrix logarithm. Let $\boldsymbol{X}_1, \boldsymbol{X}_2$ be any two points on the SPD manifold then the distance between them is given by

$$\delta_{\mathcal{M}}(\boldsymbol{X}_1, \boldsymbol{X}_2) = 0.5 \| \log(\boldsymbol{X}_1^{-\frac{1}{2}} \boldsymbol{X}_2 \boldsymbol{X}_1^{-\frac{1}{2}}) \|_F \tag{1}$$

There are other efficient methods to compute distance between two points on the SPD manifold (Gao et al., 2019; Dong et al., 2017b), however, their discussion is beyond the scope of our work. Other property of the Riemannian manifold of our interest is local diffeomorphism of geodesics which is a one-to-one mapping from the point on the tangent space of the manifold to the manifold (Pennec, 2020; Lackenby, 2020). To define such notions, let $\boldsymbol{X} \in \mathcal{S}_{++}^n$ be the base point and, $\boldsymbol{Y} \in \mathcal{T}_{\boldsymbol{X}} \mathcal{S}_{++}^n$, then Eq:(2) associates $\boldsymbol{Y} \in \mathcal{T}_{\boldsymbol{X}} \mathcal{S}_{++}^n$ to a point on the manifold (Pennec, 2020).

$$\exp_{\boldsymbol{X}}(\boldsymbol{Y}) = \boldsymbol{X}^{\frac{1}{2}} \exp(\boldsymbol{X}^{-\frac{1}{2}} \boldsymbol{Y} \boldsymbol{X}^{-\frac{1}{2}}) \boldsymbol{X}^{\frac{1}{2}} \in \mathcal{S}_{++}^n, \ \forall \boldsymbol{Y} \in \mathcal{T}_{\boldsymbol{X}} \tag{2}$$

Similarly, an inverse map is defined as $\log_{\boldsymbol{X}}(\boldsymbol{Z}) = \boldsymbol{X}^{\frac{1}{2}} \log(\boldsymbol{X}^{-\frac{1}{2}} \boldsymbol{Z} \boldsymbol{X}^{-\frac{1}{2}}) \boldsymbol{X}^{\frac{1}{2}} \in \mathcal{T}_{\boldsymbol{X}}, \ \forall \boldsymbol{Z} \in \mathcal{S}_{++}^n$.

**1) Basic operations of SPD Network:** It is well-known that operations such as mean centralization, normalization, and adding bias to a batch of data are inherent performance booster for most neural networks. In the same spirit, existing works like Brooks et al. (2019); Chakraborty (2020) use the notion of these operations for the SPD or general manifold data to define analogous operations on manifolds. Below we introduce them following the work of Brooks et al. (2019).

- *Batch mean, centering and bias*: Given a batch of $N$ SPD matrices $\{\boldsymbol{X}_i\}_{i=1}^N$, we can compute its Riemannian barycenter ($\mathscr{B}$) as $\mathscr{B} = \underset{\boldsymbol{X}_\mu \in \mathcal{S}_{++}^n}{\operatorname{argmin}} \sum_{i=1}^N \delta_{\mathcal{M}}^2(\boldsymbol{X}_i, \boldsymbol{X}_\mu)$. It is sometimes referred as Fréchet mean (Moakher, 2005; Bhatia & Holbrook, 2006). This definition can be extended to compute the weighted Riemannian Barycenter [1] also known as weighted Fréchet Mean (wFM) .

$$\mathscr{B} = \underset{\boldsymbol{X}_\mu \in \mathcal{S}_{++}^n}{\operatorname{argmin}} \sum_{i=1}^N w_i \delta_{\mathcal{M}}^2(\boldsymbol{X}_i, \boldsymbol{X}_\mu); \ \text{s.t. } w_i \geq 0 \text{ and } \sum_{i=1}^N w_i = 1 \tag{3}$$

Eq:(3) can be approximated using Karcher flow (Karcher, 1977; Bonnabel, 2013; Brooks et al., 2019) or recursive geodesic mean (Cheng et al., 2016; Chakraborty et al., 2020).

**2) Basic layers of SPD Network:** Analogous to standard CNN, methods like Huang & Van Gool (2017); Brooks et al. (2019); Chakraborty et al. (2020) designed SPD layers to perform operations that respect SPD manifold constraints. Assuming $\boldsymbol{X}_{k-1} \in \mathcal{S}_{++}^n$ be the input SPD matix to the $k^{th}$ layer, the SPD network layers are defined as follows:

- *BiMap layer*: This layer corresponds to a dense layer for SPD data. The BiMap layer reduces the dimension of a input SPD matrix via a transformation matrix $\boldsymbol{W}_k$ as $\boldsymbol{X}_k = \boldsymbol{W}_k \boldsymbol{X}_{k-1} \boldsymbol{W}_k^T$. To ensure the matrix $\boldsymbol{X}_k$ to be an SPD matrix, the $\boldsymbol{W}_k$ matrix must be of full row-rank.

- *Batch normalization layer:* To perform batch normalization after each BiMap layer, we first compute the Riemannian barycenter of the batch of SPD matrices followed by a running mean update step, which is Riemannian weighted average between the batch mean and the current running mean, with the weights $(1 - \theta)$ and $(\theta)$ respectively. Once mean is calculated, we centralize and add bias to each SPD sample of the batch using Eq:(4) (Brooks et al., 2019), where $\mathscr{P}$ is the notation used for parallel transport :

  $$\begin{aligned} &\textit{Batch centering}: \text{Centering the } \mathscr{B}: \boldsymbol{X}_i^c = \mathscr{P}_{\mathscr{B} \to I}(\boldsymbol{X}_i) = \mathscr{B}^{-\frac{1}{2}} \boldsymbol{X}_i \mathscr{B}^{-\frac{1}{2}}, I \text{ is the identity matrix} \\ &\textit{Bias the batch}: \text{Bias towards } \boldsymbol{G}: \boldsymbol{X}_i^b = \mathscr{P}_{I \to \boldsymbol{G}}(\boldsymbol{X}_i^c) = \boldsymbol{G}^{\frac{1}{2}} \boldsymbol{X}_i^c \boldsymbol{G}^{\frac{1}{2}}, I \text{ is the identity matrix} \end{aligned} \tag{4}$$

- *ReEig layer*: The ReEig layer is analogous to ReLU like layers present in the classical ConvNets. It aims to introduce non-linearity to SPD network. The ReEig for the $k^{th}$ layer is defined as: $\boldsymbol{X}_k = \boldsymbol{U}_{k-1} \max(\epsilon \boldsymbol{I}, \Sigma_{k-1}) \boldsymbol{U}_{k-1}^T$ where, $\boldsymbol{X}_{k-1} = \boldsymbol{U}_{k-1} \Sigma_{k-1} \boldsymbol{U}_{k-1}^T$, $\boldsymbol{I}$ is the identity matrix, and $\epsilon > 0$ is a rectification threshold value. $U_{k-1}, \Sigma_{k-1}$ are the orthonormal matrix and singular-value matrix respectively which are obtained via matrix factorization of $\boldsymbol{X}_{k-1}$.

---

[1] Following (Tuzel et al. (2006; 2008); Brooks et al. (2019)), we focus on the estimate of wFM with Karcher flow, and the thorough study on the general wFM's existence and uniqueness is beyond the focus of this paper.

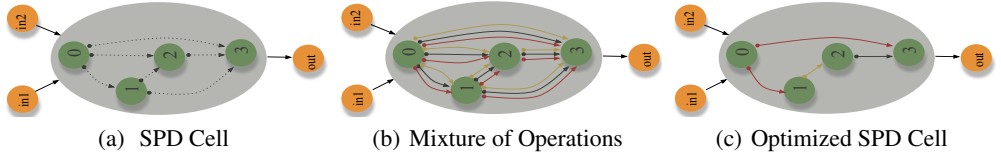

|  | (a)  SPD Cell | (b)  Mixture of Operations | (c)  Optimized SPD Cell |

Figure 1: (a) A SPD cell structure composed of 4 SPD nodes, 2 input node and 1 output node. Initially the edges are unknown (b) Mixture of candidate SPD operations between nodes (c) Optimal cell architecture obtained after solving the relaxed continuous search space under a bi-level optimization formulation.

- *LogEig layer*: To map the manifold representation of SPD to flat space so that a Euclidean operation can be performed, LogEig layer is introduced. The LogEig layer is defined as: $\boldsymbol{X}_k = \boldsymbol{U}_{k-1} \log(\Sigma_{k-1}) \boldsymbol{U}_{k-1}^T$ where, $\boldsymbol{X}_{k-1} = \boldsymbol{U}_{k-1} \Sigma_{k-1} \boldsymbol{U}_{k-1}^T$. The LogEig layer is used with fully connected layers to solve tasks with SPD representation.

- *ExpEig layer*: This layer maps the corresponding SPD representation from flat space back to SPD manifold space. It is defined as $\boldsymbol{X}_k = \boldsymbol{U}_{k-1} \exp(\Sigma_{k-1}) \boldsymbol{U}_{k-1}^T$ where, $\boldsymbol{X}_{k-1} = \boldsymbol{U}_{k-1} \Sigma_{k-1} \boldsymbol{U}_{k-1}^T$.

- *Weighted Riemannian pooling layer*: It uses wFM definition to compute the output of the layer. Recent method use recursive geodesic mean algorithm to calculate the mean (Chakraborty et al., 2020), in contrast, we use Karcher flow algorithm to compute it (Karcher, 1977) as it is simple and widely used in practice.

## 3    Neural Architecture Search of SPD Manifold Network

As alluded before, to solve the suggested problem, there are a few key changes that must be introduced. Firstly, a new definition of the computation cell is required. In contrast to the computational cells designed by regular NAS algorithms like Liu et al. (2018b); Chu et al. (2020), our computational cell —which we call as SPD cell, additionally incorporate the notion of SPD manifold geometry so that SPD representations can be treated properly. On the other hand, like the regular NAS cell design, our SPD cell can either be a **normal cell** that returns SPD feature maps of the same width and height or, a **reduction cell** in which the SPD feature maps are reduced by a certain factor in width and height. Secondly, to solve our new NAS problem will require an appropriate and diverse SPD *search space* that can help NAS method to optimize for an effective SPD cell, which can then be stacked and trained to build an efficient SPD neural network architecture.

Concretely, a SPD cell is modeled by a directed asyclic graph (DAG) which is composed of nodes and edges. In our DAG each *node* is an *latent representation* of the SPD manifold valued data i.e. an intermediate SPD feature map and, each *edge* corresponds to a *valid candidate operation* on SPD manifold (see Fig.1(a)). Each edge of a SPD cell is associated with a set of candidate SPD manifold operations ($\mathcal{O}_{\mathcal{M}}$) that transforms the SPD valued latent representation from the source node (say $\boldsymbol{X}_{\mathcal{M}}^{(i)}$) to the target node (say $\boldsymbol{X}_{\mathcal{M}}^{(j)}$). We define the intermediate transformation between the nodes in our SPD cell as: $\boldsymbol{X}_{\mathcal{M}}^{(j)} = \underset{\boldsymbol{X}_{\mathcal{M}}^{(j)}}{\operatorname{argmin}} \sum_{i<j} \delta_{\mathcal{M}}^2 \left( \mathcal{O}_{\mathcal{M}}^{(i,j)} (\boldsymbol{X}_{\mathcal{M}}^{(i)}), \boldsymbol{X}_{\mathcal{M}}^{(j)} \right)$, where $\delta_{\mathcal{M}}$ denotes the geodesic distance Eq:(1). Generally, this transformation result corresponds to the unweighted Fréchet mean of the operations based on the predecessors, such that the mixture of all operations still reside on SPD manifolds. Note that our definition of SPD cell ensures that each computational graph preserves the appropriate geometric structure of the SPD manifold. Equipped with the notion of SPD cell and its intermediate transformation, we are prepared to propose our search space (§3.1) followed by the solution to our SPDNet NAS problem (§3.2) and its results (§4).

### 3.1    Search Space

Our search space consists of a set of valid SPD network operations which is defined for the supernet search. First of all, the search space includes some existing SPD operations[2], e.g., BiMap, batch normalization, ReEig, LogEig, ExpEig and weighted Riemannian pooling layers, all of which are

---

[2]Our search space can include some other exiting SPD operations like manifoldnorm Chakraborty (2020). However, a comprehensive study on them is out of our focus, which is on studying a NAS algorithm on a given promising search space.

Table 1: Search space for the proposed SPD architecture search method.

| Operation | Definition | Operation | Definition |
|---|---|---|---|
| BiMap_0 | {BiMap, Batch Normalization} | WeightedReimannPooling_normal | {wFM on SPD multiple times} |
| BiMap_1 | {BiMap, Batch Normalization, ReEig} | AveragePooling_reduced | {LogEig, AveragePooling, ExpEig} |
| BiMap_2 | {ReEig, BiMap, Batch Normalization} | MaxPooling_reduced | {LogEig, MaxPooling, ExpEig} |
| Skip_normal | {Output same as input} | Skip_reduced = {$C_{in} = BiMap(X_{in})$, $[U_{in}, D_{in}, \sim] = svd(C_{in})$; in = 1, 2}, |  |
| None_normal | {Return identity matrix} | $C_{out} = U_b D_b U_b^T$, where, $U_b = diag(U_1, U_2)$ and $D_b = diag(D_1, D_2)$ |  |

introduced in Sec.2. Though those individual operations (e.g., BiMap, LogEig, ExpEig) have been explored well by existing works, different aggregations on them are still understudied, which are essential to enrich our search space. To be specific to enrich the search space, following Liu et al. (2018b); Gong et al. (2019) traditional NAS methods, we apply the SPD batch normalization to every SPD convolution operation (i.e., BiMap), and design three variants of convolution blocks including the one without activation (i.e., ReEig), the one using post-activation and the one using pre-activation (see Table 1). In addition, we introduce **five new operations** analogous to DARTS (Liu et al., 2018b) to enrich the search space in the context of SPD networks. These are skip normal, none normal, average pooling, max pooling and skip reduced. The effect of such diverse operation choices have not been fully explored for SPD networks. All the candidate operations are illustrated in Table (1), and the definitions of the new operations are detailed as follows:

**(a) Skip normal**: It preserves the input representation and is similar to skip connection. **(b) None normal**: It corresponds to the operation that returns identity as the output i.e, the notion of zero in the SPD space. **(c) Max pooling**: Given a set of SPD matrices, max pooling operation first projects these samples to a flat space via a LogEig operation, where a standard max pooling operation is performed. Finally, an ExpEig operation is used to map the sample back to the SPD manifold. **(d) Average pooling**: Similar to Max pooling, the average pooling operation first projects the samples to the flat space using a LogEig operation, where a standard average pooling is employed. To map the sample back to SPD manifold, an ExpEig operation is used. **(e) Skip reduced**: It is similar to 'skip_normal' but in contrast, it decomposes the input into small matrices to reduces the inter-dependency between channels. Our definition of reduce operation is in line with the work of Liu et al. (2018b).

The newly introduced operations allow us to generate a more diverse discrete search space. As presented in Table 2, the randomly selected architecture (generally consisting of the newly introduced SPD operations) shows some improvement over SPDNet and SPDNetBN, both of which only contain conventional SPD operations. This establishes the effectiveness of the introduced rich search space.

## 3.2 SUPERNET SEARCH

To solve the suggested new NAS problem, one of the most promising NAS methodologies is supernet modeling. While we can resort to some other NAS methods to solve the problem like reinforcement learning based method (Zoph & Le, 2016) or evolution based algorithm (Real et al., 2019), in general, the supernet method models the architecture search problem as a one-shot training process of a single supernet that consists of all architectures. Based on the *supernet modeling*, we can search for the optimal SPD neural architecture either using parameterization of architectures or sampling of single-path architectures. In this paper, we focus on the parameterization approach that is based on the *continuous relaxation of the SPD neural architecture representation*. Such an approach allows for an efficient search of architecture using the gradient descent approach. Next, we introduce our supernet search method, followed by a solution to our proposed bi-level optimization problem. Fig.1(b) and Fig.1(c) illustrates an overview of our proposed method.

To search for an optimal SPD architecture ($\alpha$), we optimize the over parameterized supernet. In essence, it stacks the basic computation cells with the parameterized candidate operations from our search space in a one-shot search manner. The contribution of specific subnets to the supernet helps in deriving the optimal architecture from the supernet. Since the proposed operation search space is discrete in nature, we relax the explicit choice of an operation to make the search space continuous. To do so, we use wFM over all possible candidate operations. Mathematically,

$$\bar{\mathscr{O}}_{\mathcal{M}}(\boldsymbol{X}_{\mathcal{M}}) = \underset{\boldsymbol{X}_{\mathcal{M}}^{\mu}}{\operatorname{argmin}} \sum_{k=1}^{N_e} \tilde{\alpha}^k \delta_{\mathcal{M}}^2 \Big( \mathscr{O}_{\mathcal{M}}^{(k)}(\boldsymbol{X}_{\mathcal{M}}), \boldsymbol{X}_{\mathcal{M}}^{\mu} \Big); \text{ subject to: } \mathbf{1}^T \tilde{\alpha} = 1, \ 0 \le \tilde{\alpha} \le 1 \qquad (5)$$

---

**Algorithm 1:** The proposed Neural Architecture Search of SPD Manifold Nets (SPDNetNAS)

---

**Require**: Mixed Operation $\bar{\mathscr{O}}_{\mathcal{M}}$ which is parameterized by $\alpha^k$ for each edge $k \in N_e$;
**while** *not converged* **do**

> **Step1**: Update $\alpha$ (architecture) using Eq:(8) solution by satisfying an additional strict convex constraint. Note that updates on $w$ and $\tilde{w}$ (Eq:(9), Eq:(10)) should follow the gradient descent on SPD manifold;
> **Step2**: Update $w$ by solving $\nabla_w \boldsymbol{E}_{train}(w, \alpha)$; Ensure SPD manifold gradient to update $w$ (Absil et al., 2009; Huang & Van Gool, 2017; Brooks et al., 2019);

**end**
**Ensure**: Final architecture based on $\alpha$. Decide the operation at an edge $k$ using $\underset{o \in \mathscr{O}_{\mathcal{M}}}{\text{argmax}}\{\alpha_o^k\}$

---

where, $\mathscr{O}_{\mathcal{M}}^k$ is the $k^{th}$ candidate operation between nodes, $\boldsymbol{X}_{\mu}$ is the intermediate SPD manifold mean (Eq.3) and, $N_e$ denotes number of edges. We can compute wFM solution either using Karcher flow (Karcher, 1977) or recursive geodesic mean (Chakraborty et al., 2020) algorithm. Nonetheless, we adhere to Karcher flow algorithm as it is widely used to calculate wFM[3]. To impose the explicit convex constraint on $\tilde{\alpha}$, we project the solution onto the probability simplex as

$$\underset{\alpha}{\text{minimize}} \|\alpha - \tilde{\alpha}\|_2^2; \text{ subject to: } \mathbf{1}^T \alpha = 1, \ 0 \leq \alpha \leq 1 \tag{6}$$

Eq:(6) enforces the explicit constraint on the weights to supply $\alpha$ for our task and can easily be added as a convex layer in the framework (Agrawal et al., 2019). This projection is likely to reach the boundary of the simplex, in which case $\alpha$ becomes sparse (Martins & Astudillo, 2016). Optionally, softmax, sigmoid and other regularization methods can be employed to satisfy the convex constraint. However, Chu et al. (2020) has observed that the use of softmax can cause performance collapse and may lead to aggregation of skip connections. While Chu et al. (2020) suggested sigmoid can overcome the unfairness problem with softmax, it may output smoothly changed values which is hard to threshold for dropping redundant operations with non-marginal contributions to the supernet. Also, FairDARTS (Chu et al., 2020) regularization, may not preserve the summation equal to 1 constraint. Besides, Chakraborty et al. (2020) proposes recursive statistical approach to solve wFM with convex constraint, however, the definition proposed do not explicitly preserve the equality constraint and it requires re-normalization of the solution. In contrast, our approach composes of the sparsemax transformation for convex Fréchet mixture of SPD operations with the following two advantages: 1) It can preserve most of the important properties of softmax such as, it is simple to evaluate, cheaper to differentiate (Martins & Astudillo, 2016). 2) It is able to produce **sparse distributions** such that the best operation associated with each edge is more likely to make more dominant contributions to the supernet, and thus more optimal architecture can be derived (refer Figure 2(a),2(b) and §4).

From Eq:(5–6), the **mixing of operations** between nodes is determined by the weighted combination of alpha's ($\alpha^k$) and the set of operations. This relaxation makes the search space continuous and therefore, architecture search can be achieved by learning a set of alpha ($\alpha = \{\alpha^k, \forall k \in N_e\}$). To achieve our goal, we must simultaneously learn the contribution of several possible operation within all the mixed operations ($w$) and the corresponding architecture $\alpha$. Consequently, for a given $w$, we can find $\alpha$ and vice-versa resulting in the following bi-level optimization problem.

$$\underset{\alpha}{\text{minimize}} \ \boldsymbol{E}_{val}^U(w^{opt}(\alpha), \alpha); \text{ subject to: } w^{opt}(\alpha) = \underset{w}{\text{argmin}} \ \boldsymbol{E}_{train}^L(w, \alpha) \tag{7}$$

The lower-level optimization $\boldsymbol{E}_{train}^L$ corresponds to the optimal weight variable learned for a given $\alpha$ i.e., $w^{opt}(\alpha)$ using a training loss. The upper-level optimization $\boldsymbol{E}_{val}^U$ solves for the variable $\alpha$ given the optimal $w$ using a validation loss. This bi-level search method gives optimal mixture of multiple small architectures. To derive each node in the discrete architecture, we maintain top-$k$ operations i.e, with the $k^{th}$ highest weight among all the candidate operations associated with all the previous nodes.

**Bi-level Optimization:**   The bi-level optimization problem proposed in Eq:(7) is difficult to solve. Following Liu et al. (2018b) work, we approximate $w^{opt}(\alpha)$ in the upper- optimization problem to skip inner-optimization as follows:

$$\nabla_\alpha \boldsymbol{E}_{val}^U(w^{opt}(\alpha), \alpha) \approx \nabla_\alpha \boldsymbol{E}_{val}^U(w - \eta \nabla_w \boldsymbol{E}_{train}^L(w, \alpha), \alpha) \tag{8}$$

---

[3]In Appendix, we provide some comparison between Karcher flow and recursive geodesic mean method. A comprehensive study on this is actually beyond the scope of our paper.

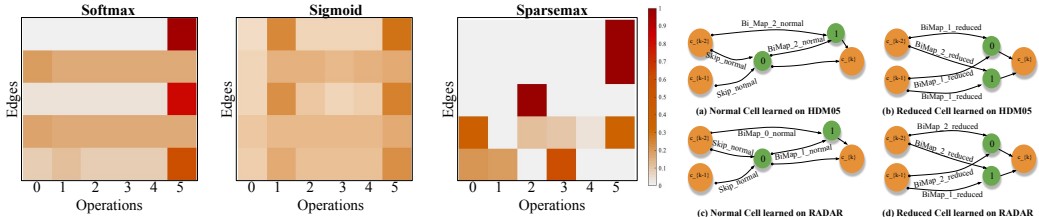

(a) Distribution of edge weights for operation selection      (b) Derived architecture

Figure 2: (a) Distribution of edge weights for operation selection using softmax, sigmoid, and sparsemax on Fréchet mixture of SPD operations. (b) Derived sparsemax architecture by the proposed SPDNetNAS. Better sparsity leads to less skips and poolings compared to those of other NAS solutions shown in Appendix Fig.5.

Here, $\eta$ is the learning rate and $\nabla$ is the gradient operator. Note that the gradient based optimization for $w$ must follow the geometry of SPD manifold to update the structured connection weight, and its corresponding SPD matrix data. Applying the chain rule to Eq:(8) gives

$$\overbrace{\nabla_\alpha \boldsymbol{E}_{val}^U(\tilde{w},\alpha)}^{\text{first term}} - \overbrace{\eta\nabla_{\alpha,w}^2 \boldsymbol{E}_{train}^L(w,\alpha)\nabla_{\tilde{w}}\boldsymbol{E}_{val}^U(\tilde{w},\alpha)}^{\text{second term}} \tag{9}$$

where, $\tilde{w} = \boldsymbol{\Psi_r}\big(w - \eta\tilde{\nabla}_w \boldsymbol{E}_{train}^L(w,\alpha)\big)$ denotes the weight update on the SPD manifold for the forward model. $\tilde{\nabla}_w, \boldsymbol{\Psi_r}$ symbolizes the Riemannian gradient and the retraction operator respectively. The second term in the Eq:(9) involves second order differentials with very high computational complexity, hence, using the finite approximation method the second term of Eq:(9) reduces to:

$$\nabla_{\alpha,w}^2 \boldsymbol{E}_{train}^L(w,\alpha)\nabla_{\tilde{w}}\boldsymbol{E}_{val}^U(\tilde{w},\alpha) = \big(\nabla_\alpha \boldsymbol{E}_{train}^L(w^+,\alpha) - \nabla_\alpha \boldsymbol{E}_{train}^L(w^-,\alpha)\big)/2\delta \tag{10}$$

where, $w^\pm = \boldsymbol{\Psi_r}(w \pm \delta\tilde{\nabla}_{\tilde{w}}\boldsymbol{E}_{val}^U(\tilde{w},\alpha))$ and $\delta$ is a small number set to $0.01/\|\nabla_{\tilde{w}}\boldsymbol{E}_{val}^U(\tilde{w},\alpha)\|_2$. Though the structure of bi-level optimization the same as the DARTS Liu et al. (2018b), there are some key differences. Firstly, the updates on the manifold-valued kernel weights are constrained on manifolds, which ensures that the feature maps at every intermediate layer are SPDs. For concrete derivations on back-propagation for SPD network layers, refer to Huang & Van Gool (2017) work. Secondly, the update on the aggregation weights of the involved SPD operations needs to satisfy an additional strict convex constraint, which is enforced as part of the optimization problem. The pseudo code of our method is outlined in **Algorithm(1)**.

## 4 EXPERIMENTS AND RESULTS

To keep the experimental evaluation consistent with the previously proposed SPD networks (Huang & Van Gool, 2017; Brooks et al., 2019), we used RADAR (Chen et al., 2006), HDM05 (Müller et al., 2007), and AFEW (Dhall et al., 2014) datasets. For SPDNetNAS, we first optimize the supernet on the training/validation sets, and then prune it with the best operation for each edge. Finally, we train the optimized architecture from scratch to document the results. For both these stages, we consider the same normal and reduction cells. A cell receives preprocessed inputs which is performed using fixed BiMap_2 to make the input of same initial dimension. All architectures are trained with a batch size of 30. Learning rate ($\eta$) for RADAR, HDM05, and AFEW dataset is set to 0.025, 0.025 and 0.05 respectively. Besides, we conducted experiments where we select architecture using a random search path (SPDNetNAS (R)), to justify whether our search space with the introduced SPD operations can derive meaningful architectures. We refer to SPDNet (Huang & Van Gool, 2017), SPDNetBN (Brooks et al., 2019), and ManifoldNet (Chakraborty et al., 2020) for comparison against handcrafted SPD networks. SPDNet and SPDNetBN are evaluated using their original implementations. We follow the video classification setup of (Chakraborty et al., 2020) to evaluate ManifoldNet on AFEW. It is non-trivial to adapt ManifoldNet to RADAR and HDM05, as ManifoldNet requires SPD features with multiple channels and both of the two datasets can hardly obtain them. For comparing against Euclidean NAS methods, we used DARTS (Liu et al., 2018b) and FairDARTS (Chu et al., 2020) by treating SPD's logarithm maps as Euclidean data in their official implementation with default setup. We observed that using raw SPD's as input to Euclidean NAS algorithms degrades its performance.

**a) Drone Recognition:** For this task, we used the RADAR dataset from (Chen et al., 2006). The synthetic setting for this dataset is composed of radar signals, where each signal is split into windows

of length 20 resulting in a 20x20 covariance matrix for each window (one radar data point). The synthesized dataset consists of 1000 data points per class. Given $20 \times 20$ input covariance matrices, our reduction cell reduces them to $10 \times 10$ matrices followed by normal cell to provide complexity to our network. Following Brooks et al. (2019), we assign 50%, 25%, and 25% of the dataset for training, validation, and test set respectively. The Euclidean NAS algorithms are evaluated on the euclidean map of the input. For direct SPD input the performance of darts(95.86%) and fairdarts (92.26%) are worse as expected. For this dataset, our algorithm takes 1 CPU day of search time to provide the SPD architecture. Training and validation take 9 CPU hours for 200 epochs[4]. Test results on this dataset are provided in Table (2) which clearly shows the benefit of our method. Statistical performance show that our NAS algorithm provides an efficient architecture with much fewer parameters (more than 140 times) than state-of-the-art Euclidean NAS on the SPD manifold valued data. The normal and reduction cells obtained on this dataset are shown in Fig. 2(b).

**b) Action Recognition:** For this task, we used the HDM05 dataset (Müller et al., 2007) which contains 130 action classes, yet, for consistency with previous work (Brooks et al., 2019), we used 117 class for performance comparison. This dataset has 3D coordinates of 31 joints per frame. Following the previous works (Harandi et al., 2017; Huang & Van Gool, 2017), we model an action for a sequence using $93 \times 93$ joint covariance matrix. The dataset has 2083 SPD matrices distributed among all 117 classes. Similar to the previous task, we split the dataset into 50%, 25%, and 25% for training, validation, and testing. Here, our reduction cell is designed to reduce the matrices dimensions from 93 to 30 for legitimate comparison against Brooks et al. (2019). To search for the best architecture, we ran our algorithm for 50 epoch (3 CPU days). Figure 2(b) show the final cell architecture that got selected based on the validation performance. The optimal architecture is trained from scratch for 100 epochs which took approximately 16 CPU hours. The test accuracy achieved on this dataset is provided in Table (2). Statistics clearly show that our models despite being lighter performs better than the NAS models and the handcrafted SPDNets. The NAS models' inferior results show that the use of SPD layers for respecting SPD geometries is crucial for SPD data analysis.

Table 2: Performance comparison of our method against existing SPDNets and TraditionalNAS on drone and action recognition. SPDNetNAS (R): randomly select architecure from our search space, DARTS/FairDARTS: accepts logarithm forms of SPDs. The **search time** of our method on RADAR and HDM05 is noted to be **1 CPU days** and **3 CPU days** respectively. And the search cost of DARTS and FairDARTS on RADAR and HDM05 are about 8 GPU hours. #RADAR and #HDM05 show model parameter comparison on the respective dataset.

| Dataset | DARTS | FairDARTS | SPDNet | SPDNetBN | SPDNetNAS (R) | SPDNetNAS |
|---------|-------|-----------|--------|----------|---------------|-----------|
| RADAR | 98.21%± 0.23 | **98.51% ± 0.09** | 93.21% ± 0.39 | 92.13% ± 0.77 | 95.49% ± 0.08 | 97.75% ± 0.30 |
| #RADAR | 2.6383 MB | 2.6614 MB | 0.0014 MB | 0.0018 MB | 0.0185 MB | 0.0184 MB |
| HDM05 | 53.93% ± 1.42 | 47.71% ± 1.46 | 61.60% ± 1.35 | 65.20% ± 1.15 | 66.92% ± 0.72 | **69.87% ± 0.31** |
| #HDM05 | 3.6800MB | 5.1353 MB | 0.1082 MB | 0.1091 MB | 1.0557 MB | 1.064MB MB |

**c) Emotion Recognition:** We used AFEW dataset (Dhall et al., 2014) to evaluate the transferability of our searched architecture for emotion recognition. This dataset has 1345 videos of facial expressions classified into 7 distinct classes. To train on the video frames directly, we stack all the handcrafted SPDNets and our searched SPDNet on top of a covolutional network Meng et al. (2019) with its official implementation. For ManifoldNet, we compute a $64 \times 64$ spatial covariance matrix for each frame on the intermediate CNN features of $64 \times 56 \times 56$ (channels, height, width). We follow the reported setup of Chakraborty et al. (2020) to first apply a single wFM layer with kernel size 5, stride 3 and 8 channels, followed by three temporal wFM layers of kernel size 3 and stride 2, with the channels being 1, 4, 8 respectively. We closely follow the official implementation of ManifoldNet [5] for the wFM layers and adapt the code to our specific task. Since SPDNet, SPDNetBN and our SPDNetNAS require a single channel SPD matrix as input, we use the final 512 dimensional vector extracted from the covolutional network, project it using a dense layer to a 100 dimensional feature vector and compute a $100 \times 100$ temporal covariance matrix. To study the transferability of our algorithm, we evaluate its searched architecture on RADAR and HDM05. In addition, we evaluate DARTS and FairDARTS directly on the video frames of AFEW. Table (3) reports the evaluations results. As we can observe, the transferred architectures can handle the new dataset quite convincingly, and their test accuracies are better than those of the existing SPDNets and the Euclidean NAS algorithms. In Appendix, we present results of competing methods and our searched models on the raw SPD features of AFEW.

---

[4]For details on the choice of CPU rather than GPU, see appendix

[5]https://github.com/jjbouza/manifold-net-vision

Table 3: Performance comparison of our transferred architectures on AFEW against handcrafted SPDNets and Euclidean NAS. SPDNetNAS(RADAR/HDM05): architectures searched on RADAR and HDM05 respectively.

| DARTS | FairDARTS | ManifoldNet | SPDNet | SPDNetBN | SPDNetNAS (RADAR) | SPDNetNAS (HDM05) |
|---|---|---|---|---|---|---|
| 26.88 % | 22.31% | 28.84% | 34.06% | 37.80% | **40.80%** | **40.64**% |

#### d) Ablation study:

Lastly, we conducted some ablation study to realize the effect of probability simplex constraint (sparsemax) on our suggested Fréchet mixture of SPD operations. Although in Fig. 2(a) we show better probability weight distribution with sparsemax, Table(4) shows that it performs better empirically as well on both RADAR and HDM05 compared to the softmax and sigmoid cases. Therefore, SPD architectures derived using the sparsemax is observed to be better.

Table 4: Ablations study on different solutions to our suggested Fréchet mixture of SPD operations.

| Dataset | softmax | sigmoid | sparsemax |
|---|---|---|---|
| RADAR | $96.47\% \pm 0.10$ | $97.70\% \pm 0.23$ | $\mathbf{97.75\% \pm 0.30}$ |
| HDM05 | $68.74\% \pm 0.93$ | $68.64\% \pm 0.09$ | $\mathbf{69.87\% \pm 0.31}$ |

**e) Statistical comparison under same model complexity:** We compare the statistical performance of our method against the other competing methods under similar model sizes. Table 5 show the results obtained on the RADAR dataset. One key point to note here is that when we increase the number of parameters in SPDNet and SPDNetBN, we observe a very severe degradation in the performance accuracy —mainly because the network starts overfitting rapidly. The performance degradation is far more severe for the HDM05 dataset with SPDNet (1.047MB) performing 0.7619% and SPDNetBN (1.082MB) performing 1.45% and hence, is not reported in the table below. That further indicates the ability of SPDNetNAS to generalize better and avoid overfitting despite the larger model size.

Similarly, we experimented on the AFEW dataset. To have a fair comparison against the related method like ManifoldNet, whose model size is about (76MB), we must reduce the model size accordingly. ManifoldNet model size is large mainly due to multiple final dense fully connected layers. Hence, to reduce the model size, we decreased the number of FC layers. The performance result with comparable model sizes on the AFEW dataset is shown in Table 5. Again, we can infer that our SPDNetNAS achieves a significant performance improvement over the others.

Table 5: Performance of our model against ManifoldNet, SPDNet and SPDNetBN with comparable model sizes on the RADAR and AFEW datasets.

| Dataset | Manifoldnet | SPDNet | SPDNetBN | SPDNetNAS |
|---|---|---|---|---|
| RADAR | NA | 73.066% | 87.866% | **97.75%** |
| #RADAR | NA | 0.01838 MB | 0.01838 MB | 0.01840 MB |
| AFEW | 25.8% | 34.06% | 37.80% | **40.64%** |
| #AFEW | 11.6476 MB | 11.2626 MB | 11.2651 MB | 11.7601 MB |

## 5 CONCLUSION AND FUTURE DIRECTION

In this work, we present a neural architecture search problem of SPD manifold networks. To solve it, a SPD cell representation and corresponding candidate operation search space is introduced. A parameterized supernet search method is employed to explore the relaxed continuous SPD search space following a bi-level optimization problem with probability simplex constraint for effective SPD network design. The solution to our proposed problem using back-propagation is carefully crafted, so that, the weight updates follow the geometry of the SPD manifold. Quantitative results on the benchmark dataset show a commendable performance gain over handcrafted SPD networks and Euclidean NAS algorithms. Additionally, we demonstrate that the learned SPD architecture is much lighter than other NAS based architecture and, it is transferable to other datasets as well.

Our work provides an architecture search methodology for the scenarios where the acquired input data are SPD's, for example, diffusion tensor imaging for medical applications, drone recognition, etc. In addition, our method offers a paradigm to automate the neural architecture design for the scenarios that require the second-order representations/poolings for robust visual recognition (e.g., Wang et al. (2017); Engin et al. (2018); Wang et al. (2019)). Accordingly, we encourage more future works to pursue these two directions. Also, it is fairly interesting to extend our proposed method to sequential manifold valued data (Zhen et al., 2019; Chakraborty et al., 2018).

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

# A ADDITIONAL EXPERIMENTAL ANALYSIS

## A.1 EFFECT OF MODIFYING PREPROCESSING LAYERS FOR MULTIPLE DIMENSIONALITY REDUCTION

Unlike Huang & Van Gool (2017) work on the SPD network, where multiple transformation matrices are applied at multiple layers to reduce the dimension of the input data, our reduction cell presented in the main paper is one step. For example: For HDM05 dataset (Müller et al., 2007), the author's of SPDNet (Huang & Van Gool, 2017) apply $93 \times 70$, $70 \times 50$, $50 \times 30$, transformation matrices to reduce the dimension of the input matrix, on the contrary, we reduce the dimension in one step from 93 to 30 which is inline with Brooks et al. (2019) work.

To study the behaviour of our method under multiple dimesionality reduction pipeline on HDM05, we use the preprocessing layers to perform dimensionality reduction. To be precise, we consider a **preprocessing** step to reduce the dimension from 93 to 70 to 50 and then, a **reduction cell** that reduced the dimension from 50 to 24. This modification has the advantage that it reduces the search time from 3 CPU days to 2.5 CPU days, and in addition, provides a performance gain (see Table (6)). The normal and the reduction cells for the multiple dimension reduction are shown in Figure (3).

Table 6: Results of modifying preprocessing layers for multiple dimentionality reduction on HDM05

| Preprocess dim reduction | Cell dim reduction | SPDNetNAS | Search time |
|---|---|---|---|
| NA | $93 \rightarrow 30$ | $68.74\% \pm 0.93$ | 3 CPU days |
| $93 \rightarrow 70 \rightarrow 50$ | $50 \rightarrow 24$ | **69.41 % $\pm$ 0.13** | 2.5 CPU days |

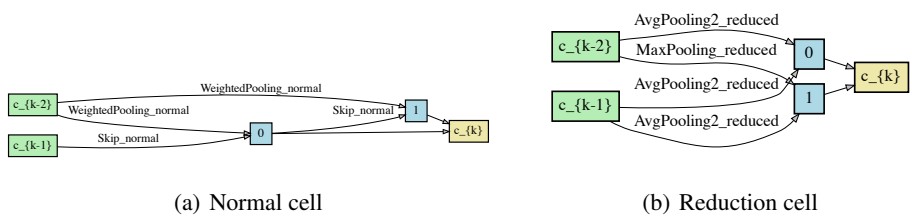

(a) Normal cell  (b) Reduction cell

Figure 3: (a)-(b) Normal cell and Reduction cell for multiple dimensionality reduction respectively

## A.2 EFFECT OF ADDING NODES TO THE CELL

Experiments presented in the main paper consists of $N = 5$ nodes per cell which includes two input nodes, one output node, and two intermediate nodes. To do further analysis of our design choice, we added nodes to the cell. Such analysis can help us study the critical behaviour of our cell design i.e, whether adding an intermediate nodes can improve the performance or not?, and how it affects the computational complexity of our algorithm? To perform this experimental analysis, we used HDM05 dataset (Müller et al., 2007). We added one extra intermediate node ($N = 6$) to the cell design. We observe that we converge towards an architecture design that is very much similar in terms of operations (see Figure 4). The evaluation results shown in Table (7) help us to deduce that adding more intermediate nodes increases the number of channels for output node, subsequently leading to increased complexity and almost double the computation time.

Table 7: Results for multi-node experiments on HDM05

| Number of nodes | SPDNetNAS | Search time |
|---|---|---|
| 5 | **68.74% $\pm$ 0.93** | 3 CPU days |
| 6 | 67.96% $\pm$ 0.67 | 6 CPU days |

## A.3 EFFECT OF ADDING MULTIPLE CELLS

In our paper we stack 1 normal cell over 1 reduction cell for all the experiments. For more extensive analysis of the proposed method, we conducted training experiments by stacking multiple cells which

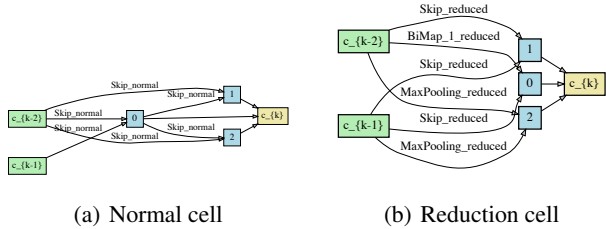

(a) Normal cell          (b) Reduction cell

Figure 4: (a)-(b) Optimal Normal cell and Reduction cell with 6 nodes on the HDM05 dataset

is in-line with the experiments conducted by Liu et al. (2018b). We then transfer the optimized architectures from the singe cell search directly to the multi-cell architectures for training. Hence, the search time for all our experiments is same as for a single cell search i.e. 3 CPU days. Results for this experiment are provided in Table 8. The first row in the table shows the performance for single cell model, while the second and third rows show the performance with multi-cell stacking. *Remarkably, by stacking multiple cells our proposed SPDNetNAS outperforms SPDNetBN Brooks et al. (2019) by a large margin (about **8%**, i.e., about **12%** for the relative improvement).*

Table 8: Results for multiple cell search and training experiments on HDM05: reduction corresponds to reduction cell and normal corresponds to the normal cell.

|  | Dim reduction in cells | Cell type sequence | SPDNetNAS | Search Time |
|---|---|---|---|---|
| single cell | $93 \rightarrow 46$ | reduction-normal | $68.74\% \pm 0.93$ | 3 CPU days |
| multi-cell | $93 \rightarrow 46$ | normal-reduction-normal | $\mathbf{71.48\% \pm 0.42}$ | 3 CPU days |
| multi-cell | $93 \rightarrow 46 \rightarrow 22$ | reduction-normal-reduction-normal | $\mathbf{73.59\ \% \pm 0.33}$ | 3 CPU days |

### A.4 AFEW PERFORMANCE COMPARISON ON RAW SPD FEATURES

In addition to the evaluation on CNN features in the major paper, we also use the raw SPD features (extracted from gray video frames) from Huang & Van Gool (2017); Brooks et al. (2019) to compare the competing methods. To be specific, each frame is normalized to $20 \times 20$ and then represent each video using a $400 \times 400$ covariance matrix (Wang et al., 2012; Huang & Van Gool, 2017). Table 9 summarizes the results. As we can see, the transferred architecture can handle the new dataset quite convincingly. The test accuracy is comparable to the best SPD network method for RADAR model transfer. For HDM05 model transfer, the test accuracy is much better than the existing SPD networks.

Table 9: Performance of transferred SPDNetNAS Network architecture in comparison to existing SPD Networks on the AFEW dataset Dhall et al. (2014). RAND symbolizes random architecture from our search space. DARTS/FairDARTS: accepts the logarithms of raw SPDs, and the other competing methods receive the SPD features.

| DARTS | FairDARTS | ManifoldNet | SPDNet | SPDNetBN | Ours(R) | Ours(RADAR) | Ours (HDM05) |
|---|---|---|---|---|---|---|---|
| 25.87 % | 25.34% | 23.98% | 33.17% | 35.22% | 32.88% | **35.31** % | **38.01%** |

### A.5 DERIVED CELL ARCHITECTURE USING SIGMOID ON FRÉCHET MIXTURE OF SPD OPERATION

Figure 5(a) and Figure 5(b) show the cell architecture obtained using the softmax and sigmoid respectively on the Fréchet mixture of SPD operation. It can be observed that it has relatively more skip and pooling operation than sparsemax ((see Figure 2(b))). In contrast to softmax and sigmoid, the SPD cell obtained using sparsemax is composed of more convolution type operation in the architecture, which in fact is important for better representation of the data.

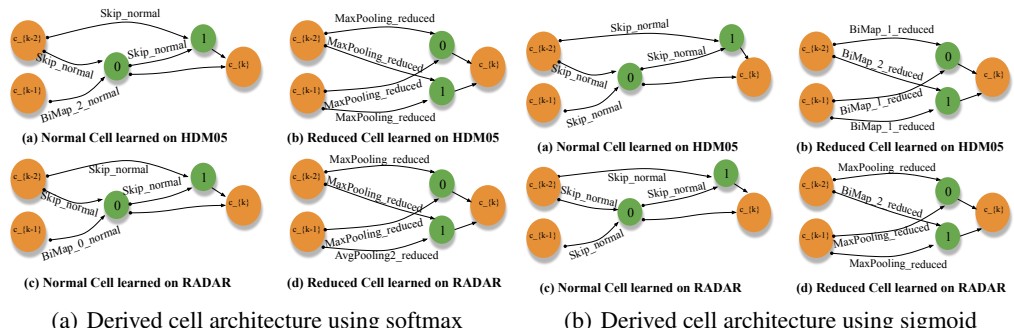

(a) Derived cell architecture using softmax      (b) Derived cell architecture using sigmoid

Figure 5: (a), (b) Derived architecture by using softmax and sigmoid on the Fréchet mixture of SPD operations. These are the normal cell and reduced cell obtained on RADAR and HDM05 dataset.

## A.6 COMPARISON BETWEEN KARCHER FLOW AND RECURSIVE APPORACH FOR WEIGHTED FRECHET MEAN

The proposed NAS algorithm is based on Fréchet Mean computations. From the weighted mixture of operations between nodes to the derivation of intermediate nodes, both compute the Fréchet mean of a set of points on the SPD manifold. It is well known that there is no closed form solution when the number of input samples is bigger than 2 (Brooks et al., 2019). We can only compute an approximation using the famous Karcher flow algorithm (Brooks et al., 2019) or recursive geodesic mean (Chakraborty et al., 2020). For comparison, we replace our used Karcher flow algorithm with the recursive approach under our SPDNetNAS framework. Table 10 sumarizes the comparison between these two algorithms. We observe considerable decrease in accuracy for both the training and test set when using the recursive methods, showing that the Karcher flow algorithm favors our proposed algorithm more.

Table 10: Test performance of the proposed SPDNetNAS using the Karcher flow algorithm and the recursive algorithm to compute Fréchet means.

| Dataset / Method | Karcher flow | Recursive algorithm |
|---|---|---|
| RADAR | **96.47% $\pm$ 0.08** | 68.13% $\pm$ 0.64 |
| HDM05 | **68.74% $\pm$ 0.93** | 56.85% $\pm$ 0.17 |

## A.7 CONVERGENCE CURVE ANALYSIS

Figure 6(a) shows the validation curve which almost saturates at 200 epoch demonstrating the stability of our training process. First column bar of Figure (6(b)) show the test accuracy comparison when only 10% of the data is used for training our architecture which demonstrate the effectiveness of our algorithm. Further, we study this for our SPDNetNAS architecture by taking 10%, 33%, 80% of the data for training. Figure 6(b)) clealy show our superiority of SPDNetNAS algorithm than handcrafted SPD networks.

Figure (7(a)) and Figure (7(b)) show the convergence curve of our loss function on the RADAR and HDM05 datasets respectively. For the RADAR dataset the validation and training losses follow a similar trend and converges at 200 epochs. For the HDM05 dataset, we observe the training curve plateaus after 60 epochs, where as the validation curve takes 100 epochs to provide a stable performance. Additionally, we noticed a reasonable gap between the training loss and validation loss for the HDM05 dataset (Müller et al., 2007). A similar pattern of convergence gap between validation loss and training loss has been observed by Huang & Van Gool (2017) work.

## A.8 WHY WE PREFERRED TO SIMULATE OUR EXPERIMENTS ON CPU RATHER THAN GPU?

When dealing with SPD matrices, we need to carry out complex computations. These computations are performed to make sure that our transformed representation and corresponding operations respect the underlying manifold structure. In our study, we analyzed SPD matrices with the Affine Invariant

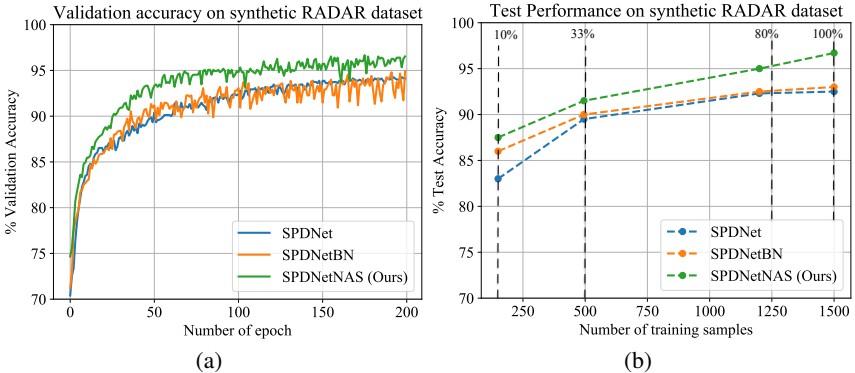

Figure 6: (a) Validation accuracy of our method in comparison to the SPDNet and SPDNetBN on RADAR dataset. Clearly, our SPDNetNAS algorithm show a steeper validation accuracy curve. (b) Test accuracy on 10%, 33%, 80%, 100% of the total data sample. It can be observed that our method exhibit superior performance.

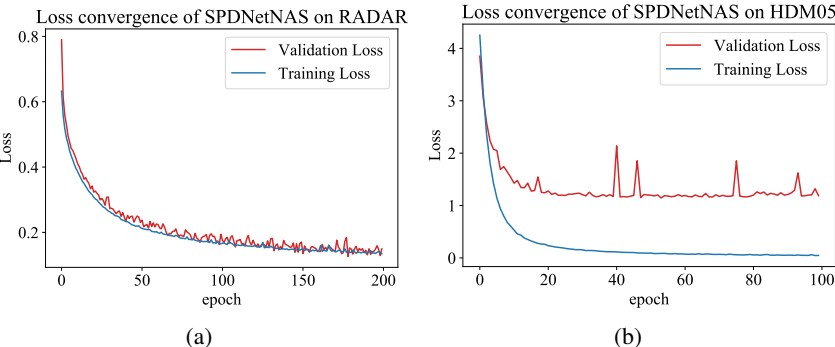

Figure 7: (a) Loss function curve showing the values over 200 epochs for the RADAR dataset (b) Loss function curve showing the values over 100 epochs on the HDM05 dataset.

Riemannian Metric (AIRM), this induces operations heavily dependent on singular value decomposition (SVD) or eigendecomposition (EIG). Both decompositions suffer from weak support on GPU platforms. Hence, our training did not benefit from GPU acceleration and we decided to train on CPU. As a future work, we aim to speedup our implementation on GPU by optimizing the SVD Householder bi-diagonalization process as studied in some existing works like Dong et al. (2017a); Gates et al. (2018).

## B  DETAILED DESCRIPTION OF OUR PROPOSED OPERATIONS

In this section, we describe some of the major operations defined in the main paper from an intuitive point of view. We particularly focus on some of the new operations that are defined for the input SPDs, i.e., the Weighted Riemannian Pooling, the Average/Max Pooling, the Skip Reduced operation and the Mixture of Operations.

### B.1  WEIGHTED RIEMANNIAN POOLING

Figure 8 provides an intuition behind the Weighted Riemannian Pooling operation. Here, w_11, w_21, etc., corresponds to the set of normalized weights for each channel (shown as two blue channels). The next channel —shown in orange, is then computed as weighted Fréchet mean over these two input channels. This procedure is repeated to achieve the desired number of output channels (here two), and finally all the output channels are concatenated. The weights are learnt as a part of the optimization procedure ensuring the explicit convex constraint is imposed.

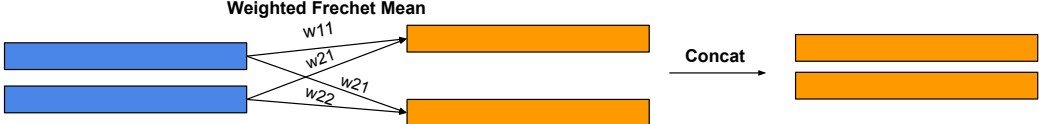

Figure 8: Weighted Riemannian Pooling: Performs multiple weighted Fréchet means on the channels of the input SPD

## B.2 Average and Max Pooling

In Figure 9 we show our average and max pooling operations. We first perform a LogEig map on the SPD matrices to project them to the Euclidean space. Next, we perform average and max pooling on these Euclidean matrices similar to classical convolutional neural networks. We further perform an ExpEig map to project the Euclidean matrices back on the SPD manifold. The diagram shown in Figure 9 is inspired by Huang & Van Gool (2017) work. The kernel size of **AveragePooling_reduced** and **MaxPooling_reduced** is set to 2 or 4 for all experiments according to the specific dimensionality reduction factors.

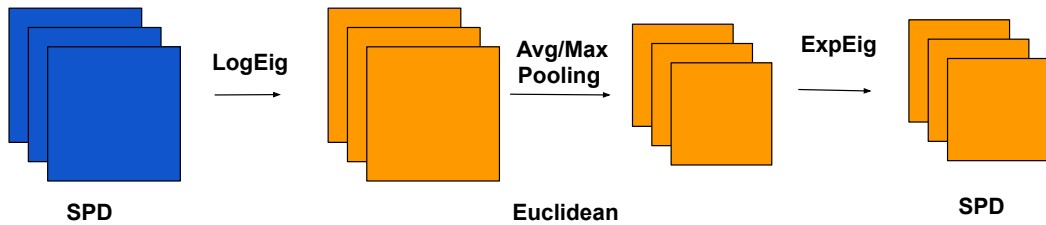

Figure 9: Avg/Max Pooling: Maps the SPD matrix to Euclidean space using LogEig mapping, does avg/max pooling followed by ExpEig map

## B.3 Skip Reduced

Following Liu et al. (2018b), we defined an analogous of Skip operation on a single channel for the reduced cell (Figure 10). We start by using a BiMap layer —equivalent to Conv in Liu et al. (2018b), to map the input channel to an SPD whose space dimension is half of the input dimension. We further perform an SVD decomposition on the two SPDs followed by concatenating the Us, Vs and Ds obtained from SVD to block diagonal matrices. Finally, we compute the output by multiplying the block diagonal U, V and D computed before.

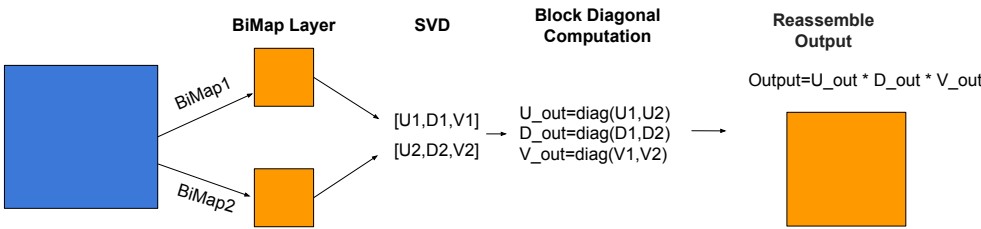

Figure 10: Skip Reduced:Maps input to two smaller matrices using BiMaps, followed by SVD decomposition on them and then computes the output using a block diagonal form of U's D's and V's

## B.4 Mixed Operation on SPDs

In Figure 11 we provide an intuition of the mixed operation we have proposed in the main paper. We consider a very simple base case of three nodes, two input nodes (1 and 2) and one output

node (node 3). The goal is to compute the output node 3 from input nodes 1 and 2. We perform a candidate set of operations on the input node, which correspond to edges between the nodes (here two for simplicity). Each operation has a weight $\alpha_{i\_j}$ where i corresponds to the node index and j is the candidate operation identifier. In Figure 11 below i and j $\in \{1, 2\}$ and $\boldsymbol{\alpha_1} = \{\alpha_{1\_1}, \alpha_{1\_2}\}$, $\boldsymbol{\alpha_2} = \{\alpha_{2\_1}, \alpha_{2\_2}\}$. $\alpha$'s are optimized as a part of the bi-level optimization procedure proposed in the main paper. Using these alpha's, we perform a channel-wise weighted Fréchet mean (wFM) as depicted in the figure below. This effectively corresponds to a mixture of the candidate operations. Note that the alpha's corresponding to all channels of a single operation are assumed to be the same. Once the weighted Fréchet means have been computed for nodes 1 and 2, we perform a channel-wise concatenation on the outputs of the two nodes, effectively doubling the number of channels in node 3.

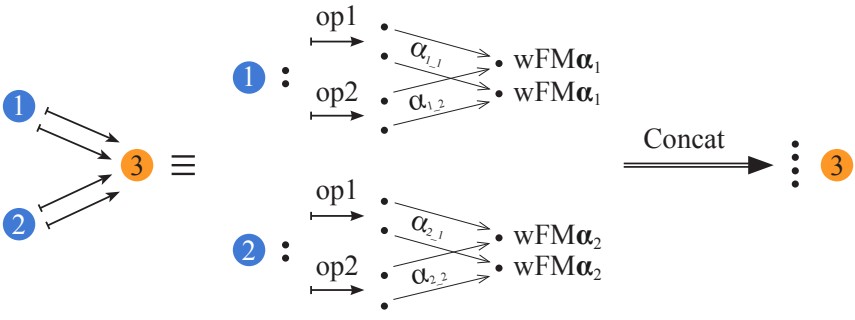

Figure 11: Detailed overview of mixed operations. We simplify the example by taking 3 nodes (two input nodes and one output node) and two candidate operations. Input nodes have two channels (SPD matrices), we perform channelwise weighted Fréchet mean between the result of each operation (edge) where weights $\alpha$'s are optimized during bi-level architecture search optimization. Output node 3 is formed by concatenating both mixed operation outputs, resulting in a four channel node.

## C    DIFFERENTIABLE CONVEX LAYER FOR SPARSEMAX OPTIMIZATION

```
1  import cvxpy as cp
2  from cvxpylayers.torch import CvxpyLayer
3
4  def sparsemax_convex_layer(x, n):
5      w_ = cp.Variable(n)
6      x_ = cp.Parameter(n)
7
8      # define the objective and constraint
9      objective = cp.Minimize(cp.sum(cp.multiply(w_, x_)))
10     constraint = [cp.sum(w_) == 1.0, 0.0 <= w_, w_<=1.0]
11
12     opt_problem = cp.Problem(objective, constraint)
13     layer = CvxpyLayer(opt_problem, parameters=[x_], variables=[w_])
14     w, = layer(x)
15     return w
```
Listing 1: Function to solve the sparsemax constraint optimization

