# OpenReview forum: "Neural Architecture Search of SPD Manifold Networks"
_ICLR.cc/2021/Conference — Reject_

### Official Review · AnonReviewer3 · 2020-10-27
**Review for Neural Architecture Search of SPD Manifold Networks**

**Rating:** 6
**Confidence:** 5

**Review:**

This paper introduces a neural architecture search method for SPD inputs, which extends DARTs from Euclidean space to SPD manifold by defining several SPD cells. Each SPD cell consists of some operations in search space of SPDNets and DARTs. The experiments are conducted on three datasets.

Strengths:
+: The idea of neural architecture search for SPD inputs seems interesting.
+: The proposed SPDNetNAS shows superiority to hand-crafted SPDNets/ManifoldNet and Euclidean NAS (i.e., DARTS and FairDARTS).
+: The paper is clear written.

Weaknesses:
-: The authors would better carefully reconsider motivation and main contribution of this paper. The current work looks like combination of SPDNets and DARTs (with some minor modifications). Then, the first attempt for NAS problem of SPD manifold networks is not an appropriate motivation for such a combination. The authors should clarify why it is necessary to search neural architecture for SPD inputs. In another words, what are drawbacks of traditional SPDNets?

For Euclidean NAS, the searched architectures aim to have lower model complexity than hand-crafted architectures, while having comparable or higher accuracies. The proposed SPDNetNAS achieves higher accuracies than hand-crafted counterparts, but has 10x times of parameters number (as shown in Table 2). Therefore, it is hard to say that performance gains come from neural architecture search or parameter increasing. Meanwhile, it is not fair for comparison of SPDNetNAS with Euclidean NAS in terms of parameters number or comparison of SPDNetNAS with SPDNets only in terms of performance.

-: The authors mainly compare with hand-crafted SPDNet and Euclidean NAS. [r1] presents a recurrent model for SPD inputs, which can handle sequential (or temporal) data, e.g., action and emotion recognition. Therefore, what are merits of the proposed SPDNetNAS over [r1] for handling sequential (or temporal) data, except architecture design vs architecture search?
[r1] A Statistical Recurrent Model on the Manifold of Symmetric Positive Definite Matrices. NIPS, 2018.

-: I wonder that why a section of FURTHER STUDY is located in appendix? Could it be integrated into section of Conclusion?

-: It is well known that two-level optimization algorithm for NAS will lead optimization gap, e.g., overfitting or model collapsing. What are strategies used to solve this issue, e.g., early stopping? If source code is not released or no more details are given, the results will be very hard to reproduce.

---

> ### Author Response · Authors · 2020-11-17
> **Response to Reviewer3**
>
> Thank you for your detailed comments. We address the concerns raised below:
>
> ### Q1: The authors would better carefully reconsider motivation and main contribution of this paper. The current work looks like combination of SPDNets and DARTs (with some minor modifications). Then, the first attempt for NAS problem of SPD manifold networks is not an appropriate motivation for such a combination. The authors should clarify why it is necessary to search neural architecture for SPD inputs. In another words, what are drawbacks of traditional SPDNets?
>
> In general, the data in fields like diffusion tensor imaging of the brain, drone imaging are collected directly as SPD samples. It is also known that non-Euclidean representation can better substantiate complex data. Thus, using traditional deep neural networks for manifold valued data can misrepresent its actual geometric structure. That is why networks like SPDNet, ManifoldNets are proposed in recent years. However, these networks are hand-designed by researchers, which has some fundamental questions to be answered.  For example, the order in which the operations/layers should be applied? Should we use a simple pooling by projection to Euclidean space or use a bimap layer?). Further, another question that could be posed is: are lighter models (in terms of parameters) efficient, or should we prefer dense models? SPDNetNAS is developed to answer some of it, if not all, of these questions.
>
> ===================================================================
>
> ### Q2: For Euclidean NAS, the searched architectures aim to have lower model complexity than hand-crafted architectures, while having comparable or higher accuracies. The proposed SPDNetNAS achieves higher accuracies than hand-crafted counterparts, but has 10x times of parameters number (as shown in Table 2). Therefore, it is hard to say that performance gains come from neural architecture search or parameter increasing. Meanwhile, it is not fair for comparison of SPDNetNAS with Euclidean NAS in terms of parameters number or comparison of SPDNetNAS with SPDNets only in terms of performance.
>
> Following your advice, we conducted an experiment on the AFEW and RADAR dataset keeping the model complexity almost similar for all the methods. The statistics show that with similar model complexity our method provides results better than other competing methods. We can add these results in the main paper.
>
> Statistical comparison on AFEW dataset under similar model complexity.
>
>  Model        |     Params(MB)     |    Accuracy
>
>   Manifoldnet   |     11.6476            |      25.8%
>
>   SPDNet         |    11.2626             |      34.06%
>
>   SPDBN          |     11.2651            |      37.80%
>
>   SPDNetNAS   |    11.7601             |     40.64%
>
> Statistical comparison on RADAR dataset under similar model complexity.
>
> Model           |     Params(MB)   |     Accuracy
>
> ManifoldNet |           NA           |      NA
>
> SPDNet        |         0.018383    |     73.066%
>
> SPDNetBN   |        0.018385    |      87.866%
>
> SPDNetNAS |        0.018400   |       97.75%
>
> (ManifoldNet uses raw images and therefore RADAR dataset is not used for its evaluation)
>
> ===================================================================
>
> ### Q3: The authors mainly compare with hand-crafted SPDNet and Euclidean NAS. [r1] presents a recurrent model for SPD inputs, which can handle sequential (or temporal) data, e.g., action and emotion recognition. Therefore, what are merits of the proposed SPDNetNAS over [r1] for handling sequential (or temporal) data, except architecture design vs architecture search? [r1] A Statistical Recurrent Model on the Manifold of Symmetric Positive Definite Matrices. NIPS, 2018.
>
> Thank you for the interesting comment. We were also aware of this work and cited it in our original submission. It is known that there are multiple deep learning techniques (e.g., convolution-based models and recurrent models) to treat video-based classification tasks. However, it is still not clear to see which one has a clear superiority. In general, the recurrent models typically suffers from gradient issues when they get deeper. Therefore, we build our NAS algorithm upon the convolution-based models (e.g., Huang et al 2017; Brooks et al. 2019; Chakraborty 2020). Nevertheless, it is still feasible to further apply our NAS algorithm to the recurrent model, and we will leave it to the future work.

---

> > ### Author Response · Authors · 2020-11-20
> > **Response to reviewer 3 continuation**
> >
> >
> >
> > ### Q4: I wonder that why a section of FURTHER STUDY is located in appendix? Could it be integrated into section of Conclusion?
> >
> > We agree to merge the future work into the conclusion of the paper.
> >
> > ===================================================================
> >
> > ### Q5: It is well known that two-level optimization algorithm for NAS will lead optimization gap, e.g., overfitting or model collapsing. What are strategies used to solve this issue, e.g., early stopping? If source code is not released or no more details are given, the results will be very hard to reproduce.
> >
> > The major problem of the issue is that the traditional DARTS uses softmax to bring an exclusive competition among candidate operations (Chu et al., 2020). To address this issue, some methods like FairDARTS use sigmoid to relax the exclusive competition of softmax. Instead,  we exploit sparsemax to make a better relaxation of softmax. In particular, it induces a better sparsity in the architecture parameters (as presented in Figure 2 (a) ) compared to softmax and sigmoid. The better sparsity generally enforces the best operations to make more dominant contributions to the superset, so that more optimal neural architecture can be achieved and the mode collapse problem (generally leading to excessive skip connections) can be overcome. Table 4 shows that the proposed sparsemax can clearly outperform softmax and sigmoid. Besides, the comparison among Fig. 2(b) and Fig.5 (a)(b) clearly shows that the proposed sparsemax can overcome the mode collapse issue (i.e., reducing the number of skip connections).  To reproduce the results, we plan to make the code public once the paper is ready for publication.

---

### Official Review · AnonReviewer1 · 2020-10-28
**The authors apply neural architecture search on SPD Manifold Networks, which is the first attempt in this field.**

**Rating:** 4
**Confidence:** 3

**Review:**

The work focus on finding better SPD Manifold Networks from the view of neural architecture search. They model the network parameterized by a set of architecture parameter and take the problem as a bi-level optimization problem, which is similar with DARTS way. I think the most contribution of this paper is the attempt to apply NAS on a new domain.

1.  The authors introduce a lot of concepts about SPD Network. But they are not meaningful in this paper since the authors just consider operations as the node. From the graph view, SPD Network can be seen as a neural architecture directly even though the operation type is different.

2. My main concern is novelty problem.  The authors also treat SPD Network as a graph in Figure 1 such that the DARTS method can be applied directly. As for Algorithm 1, they also use two types parameters: architecture weights $\alpha$  and operation weight $w$. Then bi-level optimization problem is solved using the same method with DARTS.

---

> ### Author Response · Authors · 2020-11-17
> **Response to Reviewer1**
>
> Thanks a lot for your comments. We address them below:
>
> ### Q1: The authors introduce a lot of concepts about SPD Network. But they are not meaningful in this paper since the authors just consider operations as the node. From the graph view, SPD Network can be seen as a neural architecture directly even though the operation type is different.
>
> We want to clarify this further. We first introduce the SPD network concepts because it guides the reader to understand the basic building blocks for any SPD network design. Also, we briefly discussed the essential concepts and definitions, and operations related to the SPD manifold. Note that all of these concepts, in one way or the other, are important for the SPD cell design. In our DAG representation, the operations are the edges connecting (transforming) the nodes. The nodes are the intermediate SPD feature maps and edges correspond to SPD operations.
>
> Sure SPD Network could be “one possible” candidate for the architecture search space, similar to Zoph & Le 2016. However, the cell-based approach to NAS proved to be a breakthrough in autoML; hence, we adhere to it. Furthermore, differentiable NAS methods like DARTS [1] generally fail to process SPD representations properly. This is because convolutional blocks at the core of these methods do not take the SPD manifold's rich geometry into account. Hence, the Euclidean NAS methods are seriously insufficient for SPD valued data in terms of operation choices (see their clearly inferior performances on HDM05 and AFEW in Table 2 & 3). This motivates us to define a new SPD search space and propose a novel NAS algorithm to optimize the SPD neural architectures for us.
>
> ===================================================================
>
> ### Q2:  My main concern is novelty problem. The authors also treat SPD Network as a graph in Figure 1 such that the DARTS method can be applied directly. As for Algorithm 1, they also use two types parameters: architecture weights  and operation weight . Then bi-level optimization problem is solved using the same method with DARTS.
>
> Thanks for showing your concern.  We actually used DAG (graph) to model SPD cells as shown in Fig.1. However, without respecting Riemannian geometry of SPD manifolds (or searching for SPD operations), direct application of the DARTS method is very likely to result in inferior performances. Our statistics in Table 2 and Table 3 clearly show that the direct application of DARTS and FairDARTS provides questionable results on the HDM05 and AFEW datasets. To treat the SPD representations properly, we propose a new sparsemax-based differentiable NAS method with a geometrically rich and diverse SPD neural architecture search space. The introduced search space allows us to go for a more general SPD neural architectures, and the proposed NAS method enables us to optimize the supernet of SPD neural architectures properly with our exploited sparsely weighted Frechet mean.
>
> Regarding the novelty issue in optimization, while the bi-level definition of the problem is related to DARTS, but as per the results reported in our paper, SPDNetNAS outperforms DARTS and FairDARTS significantly on HDM05 and AFEW. This is mainly because there are critical differences in our optimization method ---outlined below, that is to respect Riemannian geometry during parameter updates, which is non-trivial:
> a) The ReEig, BiMap, LogEig updates/backpropagation must be in accordance with the SPDNet. Specifically, the weights in the BiMap layers must be projected to a Steifel manifold.
> b) The Frechet mean weights must satisfy convexity constraints, which is enforced by introducing the sparsemax convex layer, contrary to just using softmax as in DARTS. We observe that using sparsemax provides clear performance gains.

---

### Official Review · AnonReviewer2 · 2020-10-29
**Incremental work using already defined tools**

**Rating:** 4
**Confidence:** 5

**Review:**

1. "In recent years, plenty of research work has been published in the area of NAS." needs citations.
2. What is "best of both these fields (NAS, SPDNets) " in page 2?
3. The definition of SPD matrix is wrong, this definition is for positive-definite matrix, not for SPD.
4. In Eq. (1), log should be matrix logarithm, and need to be mentioned.
5. In page 3, "There are other efficient methods to compute distance between two points on the SPD manifold" needs to be backed by references.
6. " Other property of the Riemannian manifoldof our interest is local diffeomorphism of geodesics which is a one-to-one mapping from the pointon the tangent space of the manifold to the manifold", this is not true, geodesic is a function of two points (start and end) and time point (say between 0 and 1), so it can not be a one-one mapping between tngent space to manifold. Authors here meant to say exponential map as defined in Eq. (2).
7. X in Eq. (2) is not the "reference point", it should be the base point for exponential map (see Chavel's book).
8. The inverse of exp map is not defined everywhere, as exp is LOCAL diffeo., please clarify.
9. Before Eq. (3), sometime->sometimes
10. "This definition is trivially extended tocompute the weighted Riemannian Barycenter" not true, the authors should mention existence and uniqueness of wFM, for example, even if FM exists, wFM may not for some choice of w.
11. What is "valid SPD" in Bimap layer, contrary to "invalid SPD"!
12. what is P in Eq. (4)? I think it is parallel transport but needs to be defined.
13. \epsilon in ReEig layer should be >0.
14. In logEig, essentially you are mapping SPD to space of symmetric matrices (which is "flat"), then why not use Inverse Exp. map? what is the necessity to define logEig map additionally?
15. Same goes for expEig map, what is need given that one can use Exp. map!
16. In weighted pooling layer, why karcher flow is "simple" and recursive method is not? Please clarify.
17. " such that the mixture of all operations still reside on SPD manifolds", here the term mixture is ambiguous!
18. Given that the operations are already defined, definition of SPD cell is NOT novel!
19. The additional search space operations are all trivial extensions of basic operations, which are not the contribution, hence I think the authors need to tone down "Most of these operations are not fully explored for SPD networks"
20. In 3.2, what the authors meant by "optimize the over parameterized supernet"?
21. In Algorithm 1, do the authors mean Riemannian gradient descent (Absil et al.)?
22. What the authors meant by "Note that the gradient based optimizationforwmust follow the geometry of SPD manifold to update the structured connection weight, and itscorresponding SPD matrix data"? Do the authors mean they need to project the Euclidean gradient found in Eq. (8) to get tangent vectors?
23. The experimental setup is weak, e.g., SPDNet and ManifoldNet need to be compared with same model complexity, otherwise the comparison is not fair!

---

> ### Author Response · Authors · 2020-11-17
> **Response to Reviewer2 Part1**
>
> Thank you for your thorough and detailed comments. We group the comments into two sections. Section 1 corresponds to major concerns with novelty/incremental work etc, and Section 2 corresponds to some minor concerns on citations/notations/typos/clarifications etc.
>
> ## Section 1: (major concerns)
>  ### Q18 and Q19: The additional search space operations are all trivial extensions of basic operations, which are not the contribution, hence I think the authors need to tone down "Most of these operations are not fully explored for SPD networks". Given that the operations are already defined, definition of SPD cell is NOT novel!
>
> Though those individual operations (e.g., BiMap, LogEig, ExpEig) have been explored well by existing works, different aggregations on them are still understudied. For example, to the best of our knowledge, conventional SPD networks rarely study some aggregations like pre-activation (i.e., BiMap_1) and skip-reduced, which are presented in Table 1. While similar concepts have been applied by regular NAS algorithms like Liu et al., 2018 and Gong et al., 2019, it is still non-trivial to design some of them like skip-reduced, which requires particular transformation and aggregation on SPD features.
>
> Moreover, enriching search space is generally essential for the success of the NAS algorithms. The newly introduced aggregation of operations allows us to enrich the discrete search space. As presented in Table 2, the randomly selected architecture (generally consisting of the newly introduced SPD operations) shows some improvement over SPDNet and SPDNetBN, both of which only contain conventional SPD operations. This reflects that the introduced new SPD operations can bring solid improvement over the traditional ones. After applying our new NAS algorithm on the enriched search space, some further improvement has been achieved. Accordingly, introducing such new aggregations to constitute a more diverse search space is regarded as the second contribution, while our most important contribution is proposing a new NAS algorithm to automate the design of SPD neural networks, which has not been done in the domain of SPD networks to our best knowledge.
>
> On the other hand, compared to the cells defined in regular NAS algorithms, our SPD cell modeling innovatively introduces essential constraints so that all the involved feature maps are enforced to reside on SPD manifolds. In this regard, our defined SPD cell is novel from existing cell definitions of the conventional NAS algorithms. Furthermore, we introduce a new NAS algorithm with sparsely weighted Frechet mean to do the optimization over the newly defined cells.
>
> However we agree that the sentence formulation above can be a bit misleading. Hence we will rephrase it in the updated version of the paper as:
> The effect of such diverse operation choices have not been fully explored for SPD networks.
>
> ================================================================================================================
>
> ### Q23: The experimental setup is weak, e.g., SPDNet and ManifoldNet need to be compared with same model complexity, otherwise the comparison is not fair!
>
> Thank you for your comment. We provide the results with comparable model sizes on RADAR and AFEW below. A key point to note here is that we observe a very severe degradation in accuracy when the number of parameters is increased in SPDNet and SPDNet BN (this is mainly because the network starts overfitting rapidly). This clearly demonstrates that simply stacking the conventional SPD layers/operations can not increase the network capacity and instead overfits the data easily. Besides, this further emphasizes the ability of SPDNetNAS to generalize better and to avoid overfitting in spite of larger model size.
>
>
> Results on RADAR under similar model complexity.
>
> Model           |     Params(MB)   |     Accuracy
>
> ManifoldNet |           NA           |      NA
>
> SPDNet        |         0.018383    |     73.066%
>
> SPDNetBN   |        0.018385    |      87.866%
>
> SPDNetNAS |        0.018400   |       97.75%
>
> (ManifoldNet uses raw images and thus RADAR is not used for its evaluation)
>
> We follow the setup of ManifoldNet as closely as possible. Its official code is for video reconstruction from[2] (https://github.com/jjbouza/manifold-net-vision) and we adapt its wFM layers from the code for our specific task of classification. For AFEW experiments the model size of ManifoldNet is about (76MB) which is mainly due to multiple dense fully connected (FC) layers. For a fairer comparison we reduce the number of FC layers. The results with comparable model sizes are as below:
>
> Results on AFEW under similar model complexity.
>
> Model        |     Params(MB)     |    Accuracy
>
> ManifoldNet   |     11.6476            |      25.8%
>
> SPDNet         |    11.2626             |      34.06%
>
> SPDBN          |     11.2651            |      37.80%
>
> SPDNetNAS   |    11.7601             |     40.64%

---

> > ### Author Response · Authors · 2020-11-17
> > **Response to Reviewer 2 (Part 2)**
> >
> > ## Section 2 (minor concerns)
> >
> > ### Q1: In recent years, plenty of research work has been published in the area of NAS. needs citations.
> > We will add the necessary citations in the revision.
> >
> > ====================================================================
> > ###  Q2: What is"best of both these fields(NAS,SPDNets)"in page2?
> > To clarify this sentence we will rewrite:
> > To automate the process of SPD network design, in this work, we take the best of both these fields (NAS, SPDNets) and propose a
> > SPD network NAS algorithm
> > as
> > To automate the design of SPD neural architectures, in this work, we choose the most promising approaches from these fields (NAS, SPD networks) and propose a novel NAS algorithm for SPD inputs.
> >
> > ====================================================================
> > ### Q3: The definition of SPD matrix is wrong, this definition is for positive-definite matrix,not for SPD.
> >
> > We think the concern raised is with respect to this sentence:
> > A real SPD matrix $\boldsymbol{X}\in\mathcal{S}^n_{++}$ satisfies the property that for any non-zero $z \in \mathbb{R}^n$, $z^T \boldsymbol{X} z>0$.
> >
> > We thank the reviewer for detailed inspection of the paper. We emphasize here that the     sentence is not the definition of an SPD matrix but merely states one of the properties of the SPD matrix as stated in [1]. Further since we already state “A real SPD” the matrix under consideration is already symmetric (in addition to property emphasized above). We refer the paper [1] for the definition of this property.
> >
> > ====================================================================
> > ### Q4: In Eq. (1), log should be matrix logarithm, and need to be mentioned
> > Thank you for the comment. We will clarify the notation in the updated version of the paper.
> >
> > ====================================================================
> > ### Q5:  In page 3, "There are other efficient methods to compute distance between two points on the SPD manifold" needs to be backed by references.
> > We will add the necessary citations in the revision.
> >
> > ====================================================================
> > ### Q6: " Other property of the Riemannian manifold of our interest is local diffeomorphism of geodesics which is a one-to-one mapping from the point on the tangent space of the manifold to the manifold", this is not true, geodesic is a function of two points (start and end) and time point (say between 0 and 1), so it can not be a one-one mapping between tangent space to manifold. Authors here meant to say exponential map as defined in Eq. (2).
> > Yes, we meant here Eq:(2). We followed Xavier Pennec, "Manifold-valued image processing with SPD matrices" Sec. 3.3.2.5. and Marc Lackenby 2020 work to present the property. In the revision, we will add this clarification.
> >
> > ==================================================================
> > ### Q7:   X in Eq. (2) is not the "reference point", it should be the base point for exponential map (see Chavel's book).
> >
> > We will make the necessary changes in the updated document. Thanks for making us aware about the rigors of topological manifolds. However, in our defense, we would like to address that the notion of reference point for defining Eq:(2) is taken from Brooks et.al. NeurIPS 2019 subsection 2.1.
> >
> > ====================================================================
> >
> > ### Q8: The inverse of exp map is not defined everywhere, as exp is LOCAL diffeo., please clarify.
> >
> > Yes, the inverse of the exp map is not defined unless exp is diffeomorphism. To add, the exp map is locally diffeomorphism w.r.t to the base point.  On SPD manifolds, the exponential map is often used as the retraction operation, and it is shown that the retraction on Riemannian manifolds is a smooth mapping from the tangent space onto the manifold with a local rigidity (local isometry) condition.  For more discussion on this, please refer to Absil et.al 2008 "Optimization algorithms on matrix manifolds" (Sec. 4.1) or Marc Lackenby, Introductory chapter on Riemannian manifolds Notes, 2020.  Thanks for showing interest in the rigors of topological manifolds and helping us improve the correctness of our draft.
> >
> > ====================================================================
> > ### Q9:  Before Eq. (3), sometime->sometimes:
> > We will make the correction.
> >
> > ====================================================================
> > ### Q10: "This definition is trivially extended to compute the weighted Riemannian Barycenter" not true, the authors should mention existence and uniqueness of wFM, for example, even if FM exists, wFM may not for some choice of w.
> > We will follow your suggestion to tune down the tone as “ This definition can be extended to compute the weighted Riemannian Barycenter”, with a footnote being "Following (Tuzel et al., 2006; 2008; Brooks et al., 2019), we focus on the approximation of wFM with Karcher flow, and the thorough study on the existence and uniqueness of the general wFM is beyond the focus of this paper."

---

> > > ### Author Response · Authors · 2020-11-17
> > > **Section 2 continuation**
> > >
> > > ====================================================================
> > > ### Q11:  What is "valid SPD" in Bimap layer, contrary to "invalid SPD"!
> > > We agree that we might have over-emphasized our point here. To address this we have removed the word “valid” ahead of SPD. We also use: valid SPD network operations. By this we mean that the operation maps an input SPD matrix to another SPD matrix.  Valid means that we expect all the feature maps respect the geometry of the SPD manifold.
> > >
> > > ====================================================================
> > > ### Q12: what is P in Eq. (4)? I think it is parallel transport but needs to be defined.
> > > We agree that the notation here was not clear enough. We will clarify it in the updated version of the paper.
> > >
> > > ====================================================================
> > > ### Q13: $\epsilon$ in ReEig layer should be >0.
> > > We will add this to the paper (we borrow the exact definition of ReEig from SPDNet:Huang and Gool 2017)
> > >
> > > ====================================================================
> > >  ### Q14 and 15:  Use of Logeig and Expeig map
> > >
> > > LogEig map has been used in works like SPDNet:Huang and Gool 2017 and SPDNetBN: Brooks et al. 2019. Though we agree that Exp. and Log. Map can be used, a more comprehensive study on them is beyond the scope of our work.
> > >
> > > ====================================================================
> > >  ### Q16: In weighted pooling layer, why karcher flow is "simple" and recursive method is not? Please clarify.
> > >
> > > Firstly Karcher flow [5] was proposed in 1977, and it is much more widely used and well-studied than the recursive method. Secondly, K in the Karcher flow algorithm as depicted in (Brooks, 2019) can be set to just 1 iteration (with no apparent accuracy degradation). This is the setup used in our experiments and is computationally inexpensive. That’s said it can be easily replaced with other methods to compute means on the SPD manifold.
> > >
> > > ===================================================================
> > > ### Q 17: "such that the mixture of all operations still reside on SPD manifolds", here the term mixture is ambiguous!
> > >
> > > We agree that the term could be ambiguous. We use the term mixture to represent a weighted combination/Frechet mean of SPD outputs of the operations/edges in the DAG. The word mixture of operation in used autoML and we borrowed it from DARTS to convey the similar underlyings.
> > >
> > > ===================================================================
> > > ### Q 20: In 3.2, what the authors meant by "optimize the over parameterized supernet"?
> > >
> > > Here, we use supernet in quite a general sense. A supernet basically contains of 2 phases (refer figure 2 in [3] for an overview)
> > > Search over different architecture types in architecture search space (overparameterized)
> > > Train the final model (the best selected architecture). Note that by definition a supernet is overparameterized as we “keep” or “preserve” all the possible architecture choices in the search phase. In the training phase we pick the optimal architecture from the search phase. Thus the search phase in a NAS algorithm or Supernet involved optimizing the overparameterized superset
> > >
> > > ===================================================================
> > > ### Q 21: In Algorithm 1, do the authors mean Riemannian gradient descent (Absil et al.)?
> > >
> > > Yes, since we follow the Huang et.al, 2017 AAAI work [4] closely for SPD optimization (which uses Absil et.al work), we unfortunately missed its reference, but we will add it in the main paper.
> > >
> > > ===================================================================
> > > ### Q 22: What the authors meant by "Note that the gradient based optimization must follow the geometry of SPD manifold to update the structured connection weight, and its corresponding SPD matrix data"? Do the authors mean they need to project the Euclidean gradient found in Eq. (8) to get tangent vectors?
> > >
> > > We would like to draw key differences in the optimization procedure of DARTS and SPDNetNAS:
> > > The weights in BiMap must lie on a Steifel manifold. Precisely, we need to enforce this constraint. Further the backpropagation for ReEig, LogEig layers also needs to be defined to respect the SPD geometry in the network. For a detailed study of the backpropagation for these layers, we request the reviewer to go through SPDNet paper [4].
> > > Secondly the convexity constraint on the architecture parameters/wFM weights needs to be enforced.

---

> > > > ### Author Response · Authors · 2020-11-20
> > > > **Section 2 continuation (citations)**
> > > >
> > > > ===================================================================
> > > >
> > > >
> > > > [1] Harandi, M., Salzmann, M. and Hartley, R., 2017. Dimensionality reduction on SPD manifolds: The emergence of geometry-aware methods. IEEE transactions on pattern analysis and machine intelligence, 40(1), pp.
> > > >
> > > > [2] Chakraborty, R., Bouza, J., Manton, J. and Vemuri, B.C., 2018. Manifoldnet: A deep network framework for manifold-valued data. arXiv preprint arXiv:1809.06211.
> > > >
> > > > [3] Yu, K., Ranftl, R. and Salzmann, M., 2020. How to Train Your Super-Net: An Analysis of Training Heuristics in Weight-Sharing NAS. arXiv preprint arXiv:2003.04276.
> > > >
> > > > [4] Huang, Z. and Van Gool, L., 2016. A riemannian network for spd matrix learning. arXiv preprint arXiv:1608.04233.
> > > >
> > > > [5] Karcher, H., 1977. Riemannian center of mass and mollifier smoothing. Communications on pure and applied mathematics, 30(5), pp.509-541.

---

### Official Review · AnonReviewer4 · 2020-10-30
**Interesting idea, however, more explanations and comparisons needed.**

**Rating:** 7
**Confidence:** 4

**Review:**

The paper considers a generalization of convolutional neural networks (CNNs) to manifold-valued data such as Symmetric Positive Definite (SPD). This paper proposes a neural architecture search problem of SPD manifold networks. A SPD cell representation and corresponding candidate operation search space is introduced. They demonstrate on drone, action and emotion recognition datasets that their method is performed well compared to SPD approaches.

- It is not clear how the computational graph preserves the geometric structure of SPD manifold.
- It is unclear how to obtain the weights of Frechet Mean w_i of wFM (eq. 3) by backpropagation.
- Relatively thorough experimental valuations: using 3 datasets comparing with sufficient number of prior
approaches. However, the comparisons of the experiments results are limited to the SPD methods.  It is hard to understand how the generalization of convolutional neural networks (CNNs) to Symmetric Positive Definite (SPD) presented in this paper helps to improve CNN.

---

> ### Author Response · Authors · 2020-11-17
> **Response to Reviewer4**
>
> We thank the reviewer for the detailed and constructive feedback. We address each of the comments below:
> ### Q1: It is not clear how the computational graph preserves the geometric structure of SPD manifold.
>
> Generic CNN layers in a neural network are designed to operate on Euclidean data (eg: images). Therefore, the only restriction on the intermediate feature maps is that they should be a Euclidean data point. However, as observed by many research works (e.g., SPDNet [2], SPDNetBN [3], ManifoldNet [4]), these networks cannot deal with SPD valued data representation properly. Subsequently, a few SPD valued data based neural networks like SPDNet [2] and SPDNetBN [3] appeared in recent years, but they are merely hand-crafted.  Thus, in this work, a NAS algorithm is proposed to automate the neural architecture design on SPD valued data. Analogous to DARTS [1] cell design, we defined SPD cell. Contrary to [1], our SPD cell must respect the following two criteria:
> 1))The output of every “edge” or “operation” in the computational graph is a valid point on the SPD manifold. We ensure this by our search space design and optimization procedure (ie. the design of operations or edges connecting the nodes).
> 2)) NAS algorithms like DARTS [1] take the weighted mean of the outputs of different operations. However, that cannot be used directly to compute the SPD matrices' mean. There are multiple approaches in the literature to compute the average of a given set of SPDs. We used the Karcher Flow [5] algorithm as it is one of the most popular methods to calculate the SPD matrices' mean. Our specific design choices ensure that every intermediate layer's output and the mixture of operations is a valid point on the spd manifold. Furthermore, we introduce sparsemax to Karcher Flow that is capable of producing sparse distribution to benefit the differentiable NAS algorithm.
>
> ================================================================================================================
>
> ### Q2:  It is unclear how to obtain the weights of Frechet Mean w_i of wFM (eq. 3) by backpropagation.
>
> The weights in the wFM (Eq. 3) finally correspond to  the architecture parameters that are optimized using the bi-level optimization. Note that by the formulation of the bi-level optimization problem the updates of the architecture parameters are dependent on updates to the network weights w. However there are key differences from DARTS [1] here:
> a) The optimization procedure of BiMap layers (network weights w), which are a crucial operation part of the SPD cell, must ensure that the weight parameters W should lie on a Stiefel Manifold. Hence the optimizers are constrained to respect this. For details on backpropagation in BiMap, LogEig and ReEig layers we refer the reviewer to SPDNet paper (Huang et.al AAAI 2017).
> b) The optimization procedure must ensure that the weights_i>0 and sum_w_i=1. For this we first use a simple softmax (which respects these constraints) and secondly we enforce the convexity constraint using a convex optimization layer for sparsemax. (We observe that using sparsemax provides clear gains). In the revision, we will add this clarification.
>
> ================================================================================================================
>
> ### Q3: Relatively thorough experimental valuations: using 3 datasets comparing with sufficient number of prior approaches. However, the comparisons of the experiments results are limited to the SPD methods. It is hard to understand how the generalization of convolutional neural networks (CNNs) to Symmetric Positive Definite (SPD) presented in this paper helps to improve CNN.
>  We clarify our experimental setup below:
> 1) To observe the benefits of using SPDNetNAS instead of DARTS that is based on traditional CNNs, we feed the Euclidean data (i.e., Log-Euclidean projection of SPD inputs) to DARTS, since DARTS is designed for Euclidean inputs. The comparison with DARTS and FairDARTS in table 2 shows the superiority of SPDNetNAS in terms of performance and model sizes.
> 2) We also conducted experiments using the SPD matrices directly as input to DARTS and we observe a clear performance degradation. This is because CNNs cannot exploit the rich manifold structure properly.
> 3) On AFEW, we apply the conventional SPDNets and ours to the top of the traditional CNNs to learn the second-order pooling over video frames for more effective emotion recognition. Compared to the competitors, this application brings clear improvement.
>
> Accordingly, the major aim of the proposed method is to provide an architecture search methodology for scenarios where the input data is inherently collected as points on an SPD manifold (e.g., medical applications, drone recognition etc) as well as those CNN scenarios that require the second-order representations/poolings.
>
> ================================================================================================================

---

> > ### Author Response · Authors · 2020-11-20
> > **Response to Reviewer4 (citations)**
> >
> > [1] Liu, H., Simonyan, K. and Yang, Y., 2018. Darts: Differentiable architecture search. arXiv preprint arXiv:1806.09055.
> >
> > [2] Huang, Z. and Van Gool, L., 2016. A riemannian network for spd matrix learning. arXiv preprint arXiv:1608.04233.
> >
> > [3] Brooks, D., Schwander, O., Barbaresco, F., Schneider, J.Y. and Cord, M., 2019. Riemannian batch normalization for SPD neural networks. In Advances in Neural Information Processing Systems (pp. 15489-15500).
> >
> > [4] Chakraborty, R., Bouza, J., Manton, J. and Vemuri, B.C., 2020. Manifoldnet: A deep neural network for manifold-valued data with applications. IEEE Transactions on Pattern Analysis and Machine Intelligence.
> >
> > [5] Karcher, H., 1977. Riemannian center of mass and mollifier smoothing. Communications on pure and applied mathematics, 30(5), pp.509-541.

---

### Author Response · Authors · 2020-11-17
**General Response to Reviewers**

We thank all the reviewers for their constructive feedback. We address the major concerns about the novelty and the significance of the proposed method below, followed by an additional comment on the summary of our major updates in the revision.

-  SPD Cell design:
Differentiable NAS methods (such as DARTS [1]) fail to process SPD inputs properly. This is because convolutional blocks at the core of these methods cannot exploit the SPD manifold's rich geometry to its fullest. Hence the Euclidean NAS methods are seriously insufficient for SPD valued data in terms of operation choices. Further, the design of networks like SPDNet [2], ManifoldNet [4] is restrictive as it is still hand-engineered. Thus the design of a richer discrete search space of SPD operations/layers becomes important to obtain more generalized SPD neural architecture. To this challenging end, we introduced some new operations to enrich the search space. Table 2 shows that the randomly selected architectures, that generally uses the new operations, have clear improvement over those only containing conventional operations. Since designing such a rich search space of SPD operations has not been attempted before, we believe this is a valuable contribution (i.e., our second important contribution).

- Relaxation of search space:
The second challenge with the NAS design for SPD inputs is to ensure that its transformation performed between nodes in the directed acyclic graph (DAG) is again SPD. For that, we used the notion of Frechet mean in SPD geometry. Note that the notion of Frechet mean on SPD manifold has not been exploited previously for continuous relaxation of NAS search space, which is essential for differentiable NAS methods. For the continuous relaxation, we need to ensure that the mixture of different operations/edges is also an SPD, which motivates using a weighted Frechet mean (wFM). However, wFM generally requires that the weights respect the constraints w_i>0 and sum_w_i=1 (note that this is often enforced using regularizers eg: ManifoldNet). As a primary choice, we use a standard softmax to ensure this constraint, but softmax cannot produce sparse distribution, and thus the softmax usage is very likely to prevent the supernet from converging to a dominant candidate architecture. To address this problem, we inovatively introduce a sparsemax optimization as a convex layer to enforce a sparsely weighted Frechet mean. This property of sparsity enables the best operations make more dominant contributions to the supernet during the search phase, and thus more optimal architecture can be derived. This contribution (i.e., our first important contribution) highly benefits NAS frameworks as studied in our evaluation on HDM05 and AFEW in Table 2 & 3. Moreover, to our best knowledge, studying NAS in the domain of SPD manifold networks is brand-new.

- Optimization:
We keep the structure of bilevel optimization the same as the DARTS [1]. However, there are some key differences. Firstly, the updates on the manifold-valued kernel weights are constrained on manifolds, which ensures that the feature maps at every intermediate layer are SPDs. Secondly, the update on the sparse aggregation weights of the involved SPD operations need to satisfy an additional strict convex constraint, which is enforced as part of the optimization problem.

- General Applicability:
It is well-known that rich data representation is the key to build robust systems. Thus, manifold valued data, manifold constraints, and manifold poolings are at the heart of action/emotion recognition, medical imaging [5], radar imaging, forensics, appearance tracking [6] to name a few. Designing neural architectures for such data representations is exceptionally time-consuming and often requires domain expertise, limiting its use to a smaller community of scientists and engineers. This work offers a good paradigm that automates the architecture design for a wide range of manifold valued representations.

[1] Liu, H., Simonyan, K. and Yang, Y., 2018. DARTS: Differentiable architecture search. arXiv preprint arXiv:1806.09055.

[2] Huang, Z. and Van Gool, L., 2016. A Riemannian network for SPD matrix learning. arXiv preprint arXiv:1608.04233.

[3] Brooks, D., Schwander, O., Barbaresco, F., Schneider, J.Y. and Cord, M., 2019. Riemannian batch normalization for SPD neural networks. In Advances in Neural Information Processing Systems.

[4] Chakraborty, R., Bouza, J., Manton, J. and Vemuri, B.C., 2020. Manifoldnet: A deep neural network for manifold-valued data with applications. IEEE Transactions on Pattern Analysis and Machine Intelligence.

[5] Gur, Y. and Sochen, N., 2009. Coordinates-Based Diffusion Over the Space of Symmetric Positive-Definite Matrices. In Visualization and Processing of Tensor Fields. Springer.

[6] Cheng, G. and Vemuri, B.C., 2013. A novel dynamic system in the space of SPD matrices with applications to appearance tracking. SIAM journal on imaging sciences.

---

> ### Author Response · Authors · 2020-11-24
> **Summary of Major Updates in the Revision**
>
> We have posted a revised version of the draft in accordance with the changes suggested by the reviewers. In a nutshell, we made the following changes:
>
> 1) Improved the motivation of the paper (Reviewer 3)
>
> 2) Polished the major novelties/contributions of the paper (Reviewer 1, Reviewer 2)
>
> 3) Clarified the meaning and notions used for the definition of an SPD Cell (Reviewer 1, Reviewer 2, Reviewer 4)
>
> 4) Added clarifications on the introduced search space (Reviewer 2)
>
> 5) Further clarified the bi-level optimization problem for SPDNetNAS (Reviewer 1, Reviewer 4)
>
> 6) Added citations and further clarified questions raised about rigors of notions in SPD geometry (Reviewer 2)
>
> 7) Reported results with raw SPD input to DARTS and FAIRDARTS on RADAR dataset (in the text) (Reviewer 4)
>
> 8) Reported results with comparable model sizes on RADAR (table), AFEW (table), HDM05 (in the text) (Reviewer 2, Reviewer 3)
>
> 9) Merged the Conclusion with the Future work section (Reviewer 3)
>
> 10) Fixed typos, added citations, clarified notations, and rephrased certain sentences which were ambiguous (Reviewer 2)

---

### Decision · Program_Chairs · 2021-01-07
**Final Decision**

**Decision:**

Reject

**Comment:**

The paper tries to find better semi-positive definite (SPD) manifold networks using neural architecture search. However, as pointed out by the reviewers, the paper has a few weaknesses: (a) it lacks in novelty, (b) it lacks in experiments that are mentioned in the SOTA papers, (c) the experiments should be performed with the same model complexity for fairness.

---

> ### Author Response · Authors · 2021-01-17
> **Clarification on comment (b) and (c)**
>
> Since the final comments do not reflect our work correctly as per the paper's content, we would like to let the readers know that, the
> statement (b) and (c) as stated by the AC are incorrect. Both of these statements are arguable after rebuttal. Based on our original submission that has compared SOTA methods with the model complexity, we added all the remaining experiments (including comparable model complexity) that reviewers asked for and presented those statistics in the rebuttal comment section and the final paper draft.